# DIFF-INSTRUCT*: TOWARDS HUMAN-PREFERRED ONE-STEP TEXT-TO-IMAGE GENERATIVE MODELS

## ABSTRACT

In this paper, we introduce the Diff-Instruct* (DI*), an image data-free approach for building one-step text-to-image generative models that align with human preference while maintaining the ability to generate highly realistic images. We frame human preference alignment as online reinforcement learning using human feedback (RLHF), where the goal is to maximize the reward function while regularizing the generator distribution to remain close to a reference diffusion process. Unlike traditional RLHF approaches, which rely on the KL divergence for regularization, we introduce a novel score-based divergence regularization, which leads to significantly better performances. Although the direct calculation of this divergence remains intractable, we demonstrate that we can efficiently compute its *gradient* by deriving an equivalent yet tractable loss function. Remarkably, with Stable Diffusion V1.5 as the reference diffusion model, DI* outperforms *all* previously leading models by a large margin. When using the 2.6B Stable Diffusion XL architecture, the DI* results in a solid human-preferred one-step model that is able to generate aesthetic images of $1024 \times 1024$ resolutions. When using the 0.6B PixelArt-$\alpha$ model as the reference diffusion, DI* achieves a new record Aesthetic Score of 6.30 and an Image Reward of 1.31 with only a single generation step, almost doubling the scores of the rest of the models with similar sizes. It also achieves an HPSv2.0 score of 28.70, establishing a new state-of-the-art benchmark, with a better layout, richer details, and aesthetic colors.

## 1 INTRODUCTIONS

Deep generative models have made substantial progress these years, largely transforming the content creation and editing across various domains (Karras et al., 2020; Nichol & Dhariwal, 2021; Poole et al., 2022; Kim et al., 2022; Tashiro et al., 2021; Meng et al., 2021; Couairon et al., 2022; Ramesh et al., 2022; Esser et al., 2024). These models demonstrated exceptional capabilities in generating high-resolution outputs, such as photorealistic images, videos, audio, and 3D assets (Oord et al., 2016; Ho et al., 2022; Poole et al., 2022; Brooks et al., 2024), and others (Zhang et al., 2023a; Xue et al., 2023; Luo & Zhang, 2024; Luo et al., 2023b; Zhang et al., 2023b; Feng et al., 2023; Deng et al., 2024; Luo et al., 2024d; Geng et al., 2024; Wang et al., 2024; Pokle et al., 2022).

Within this field, two types of generative models have gained significant attention, diffusion models and one-step generators. Diffusion models (DMs) (Sohl-Dickstein et al., 2015; Ho et al., 2020), or score-based generative models (Song et al., 2020), first progressively corrupt data with diffusion processes and then train models to approximate the score functions of the noisy data distributions across varying noise levels. The learned score functions can be used in the reverse process to generate high-quality samples by iterative denoising the noisy samples through stochastic differential equations. While DMs can produce high-quality outputs, they often require a large number of model evaluations, which limits their efficiency in applications.

Different from diffusion models, one-step generators (Zheng & Yang, 2024; Kang et al., 2023a; Sauer et al., 2023a; Yin et al., 2024; Zhou et al., 2024a; Luo et al., 2024c) have emerged as a highly efficient alternative to multi-step diffusion models. Unlike DMs, one-step generative models directly transform latent noises to output samples with a single neural network forward pass. This mechanism significantly reduced the inference cost, making them ideal for real-time applications such as text-to-image and text-to-video generations. Many existing works have demonstrated the

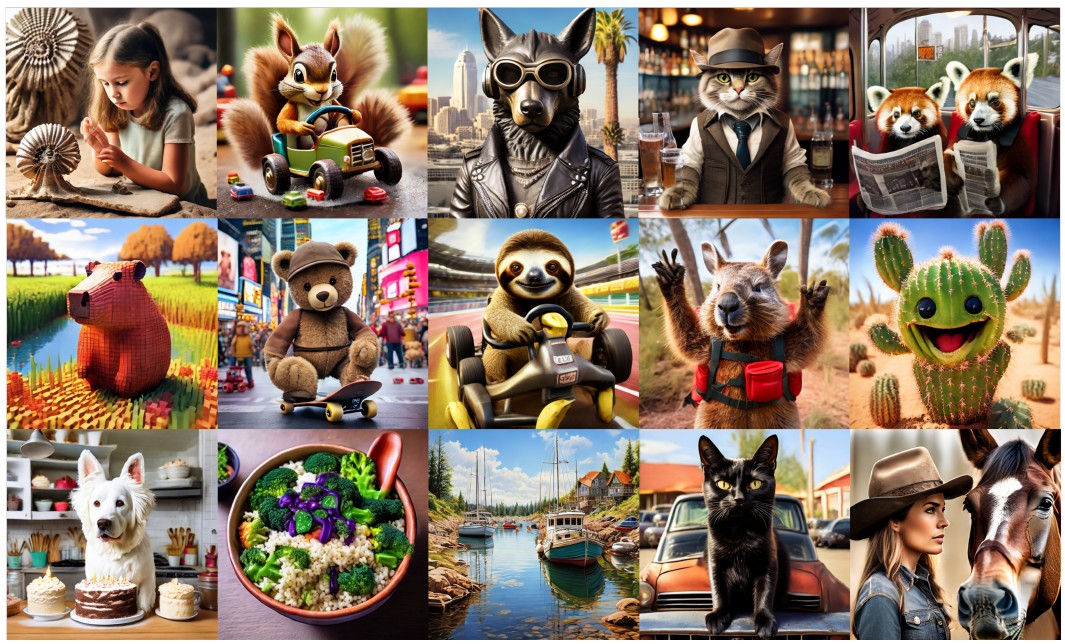

Figure 1: None cherry-picked generated images from one-step 0.6B DiT-DI* model with an record-breaking HPSv2.0 score of 28.70. After being trained with Diff-Instruct*, the images show better layouts, rich colors, vivid details, and aesthetic appearance, making them favored in terms of human preferences. Refer to the Appendix B.1 for the prompts used in comparison.

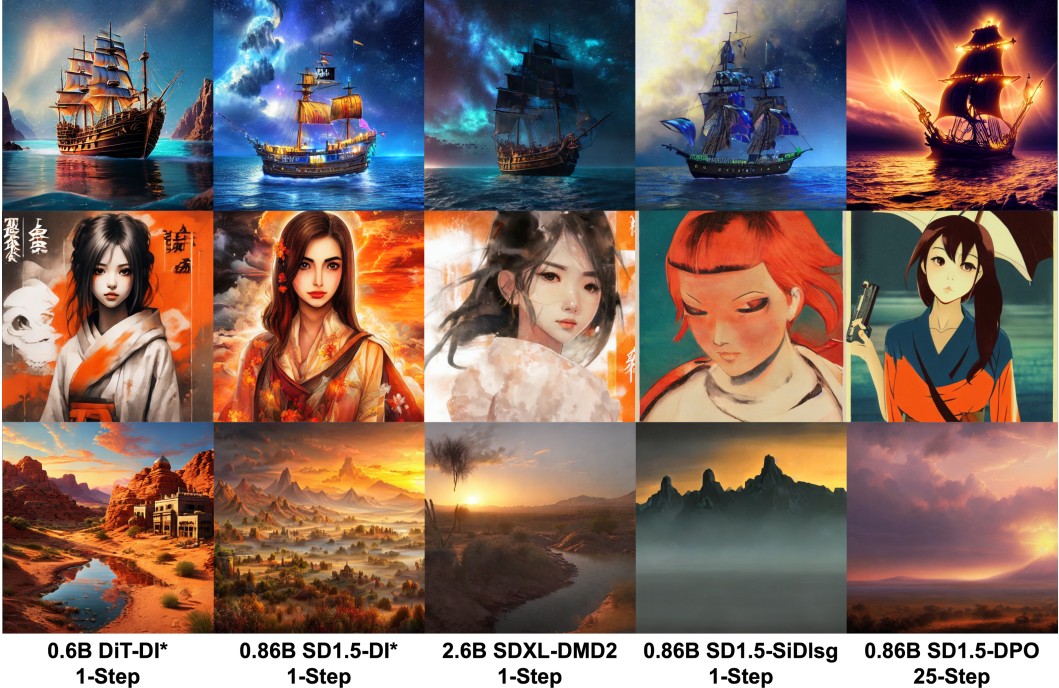

| 0.6B DiT-DI* | 0.86B SD1.5-DI* | 2.6B SDXL-DMD2 | 0.86B SD1.5-SiDlsg | 0.86B SD1.5-DPO |
| 1-Step | 1-Step | 1-Step | 1-Step | 25-Step |

Figure 2: A visual comparison of our 0.6B DiT-DI* and 0.86B SD1.5-DI* models against other text-t0-image models. Refer to the Appendix B.1 for the prompts used in comparison.

leading performances of one-step text-to-image generators (Zhou et al., 2024a; Yin et al., 2024; Luo et al., 2024c) by employing diffusion distillation (Luo, 2023). However, these works focus on matching the generator distributions with pretrained diffusion models, without considering the critical challenge of teaching one-step text-to-image models to satisfy human preferences, which is one of the most important needs in the age of human-centric AI.

To close this gap, we introduce Diff-Instruct* (DI*), a novel approach to train human-preferred one-step generators. Inspired by the success of reinforcement learning using human feedback (RLHF) in training large language models (Christiano et al., 2017; Ouyang et al., 2022), we frame the human-preference alignment problem as maximizing the rewards with a score-based divergence constraint. This yields generated samples that not only adhere to user prompts but also improve in aesthetic quality. Our approach differs from traditional RLHF methods, which rely on Kullback-Leibler (KL) divergence for distribution regularization. Instead, we propose score-based divergences, which offer more stable training dynamics and better final performance. While the direct computation of the score-based divergence is intractable, we develop a novel solution by deriving a tractable gradient computation method, leading to a tractable pseudo-loss that stands equivalent to the intractable one in the gradient perspective. With DI*, we are able to train large-scale human-preferred one-step text-to-image generative models.

In our evaluation, our best 0.6B DiT-DI*-1step model, using diffusion transformer (DiT) (Peebles & Xie, 2022) as generator architecture and PixelArt-$\alpha$ (Chen et al., 2023) as reference diffusion model, has set a new benchmark on the COCO-2017 validation prompts dataset. It achieves an ImageReward (Xu et al., 2023a) of 1.31 and an Aesthetic Score(Schuhmann, 2022) of 6.30, significantly outperforming the previous leading models such as 2.6B SDXL-DMD2 4-step models (Yin et al., 2024) and Stable Diffusion XL (Podell et al., 2023) with 25 model steps by **50.5%** and **77.0%**, respectively. Moreover, this model also reaches a new high with Human Preference Score V2.0 (Wu et al., 2023) of **28.70**, surpassing leading models like SDXL-DMD2, the PixelArt-$\alpha$, DALL-E 2 (Ramesh et al., 2022) and the Stable Diffusion XL with Refiners. When using Stable Diffusion V1.5 as the reference model and the same UNet architecture for the generator, our SD1.5-DI*-1step model with only 0.86B parameters exceeds the performance of 2.6B SDXL-DMD2 (Yin et al., 2024) and SD1.5-DPO (Wallace et al., 2024). These results establish Diff-Instruct* as a versatile, architecture-flexible approach for aligning one-step text-to-image generators with human preferences.

## 2 PRELIMINARY

**Diffusion Models.** In this section, we introduce preliminary knowledge and notations about diffusion models. Assume we observe data from the underlying distribution $q_d(\boldsymbol{x})$. The goal of generative modeling is to train models to generate new samples $\boldsymbol{x} \sim q_0(\boldsymbol{x})$. Under mild conditions, the forward diffusion process of a diffusion model can transform initial distribution $q_0$ towards some simple noise distribution,

$$\mathrm{d}\boldsymbol{x}_t = \boldsymbol{F}(\boldsymbol{x}_t, t)\mathrm{d}t + G(t)\mathrm{d}\boldsymbol{w}_t, \tag{2.1}$$

where $\boldsymbol{F}$ is a pre-defined vector-valued drift function, $G(t)$ is a pre-defined scalar-value diffusion coefficient, and $\boldsymbol{w}_t$ denotes an independent Wiener process. A continuous-indexed score network $\boldsymbol{s}_\varphi(\boldsymbol{x}, t)$ is employed to approximate marginal score functions of the forward diffusion process (2.1). The learning of score networks is achieved by minimizing a weighted denoising score matching objective (Vincent, 2011; Song et al., 2020),

$$\mathcal{L}_{DSM}(\varphi) = \int_{t=0}^{T} \lambda(t) \mathbb{E}_{\substack{\boldsymbol{x}_0 \sim q_0, \\ \boldsymbol{x}_t | \boldsymbol{x}_0 \sim p_t(\boldsymbol{x}_t | \boldsymbol{x}_0)}} \|\boldsymbol{s}_\varphi(\boldsymbol{x}_t, t) - \nabla_{\boldsymbol{x}_t} \log p_t(\boldsymbol{x}_t | \boldsymbol{x}_0)\|_2^2 \mathrm{d}t. \tag{2.2}$$

Here the weighting function $\lambda(t)$ controls the importance of the learning at different time levels and $p_t(\boldsymbol{x}_t | \boldsymbol{x}_0)$ denotes the conditional transition of the forward diffusion (2.1). After training, the score network $\boldsymbol{s}_\varphi(\boldsymbol{x}_t, t) \approx \nabla_{\boldsymbol{x}_t} \log q_t(\boldsymbol{x}_t)$ is a good approximation of the marginal score function of the diffused data distribution.

## 3 PREFERENCE ALIGNMENT USING DIFF-INSTRUCT*

In this section, we introduce Diff-Instruct*, a general method tailored for training one-step text-to-image generator according to human preference. We begin with outlining the problem setup and

defining notations. We frame the training objective as a special form of reinforcement learning using human feedback (RLHF). We then introduce a family of score-based probability divergences and show how these divergences effectively regularize the RLHF process to align closely with human aesthetics and content preferences. Next, we introduce the classifier-free reward and discuss how to balance the explicit human reward model and the implicit classifier-free reward.

**Problem Setup.** Assume we have a human reward function $r(\boldsymbol{x}_0, \boldsymbol{c})$, which encodes the human preference for an image $\boldsymbol{x}_0$ and corresponding text description $\boldsymbol{c}$. Besides, we also have a pre-trained diffusion model which will later act as a reference distribution $p_{ref}(\boldsymbol{x}_0) = q_0(\boldsymbol{x}_0)$. We assume the reference diffusion model is specified by the score function

$$\boldsymbol{s}_{q_t}(\boldsymbol{x}_t) := \nabla_{\boldsymbol{x}_t} \log q_t(\boldsymbol{x}_t) \tag{3.1}$$

where $q_t(\boldsymbol{x}_t)$'s is the underlying distribution diffused at time $t$ according to (2.1). We assume that the pre-trained diffusion model well captures the ground-truth data distribution, and thus will be the only item of consideration for our approach.

Our goal is to train a human-preferred one-step generator model $g_\theta$ that generates images by directly mapping a random noise $\boldsymbol{z} \sim p_z$ to obtain $\boldsymbol{x}_0 = g_\theta(\boldsymbol{z}, \boldsymbol{c})$, conditioned on the input text $\boldsymbol{c} \sim \mathcal{C}$. The generator's output distribution, $p_\theta(\boldsymbol{x}_0|\boldsymbol{c})$, should maximize the expected human rewards while adhering to the constraints on the divergence to the reference distribution $p_{ref}(\cdot)$. Let $\boldsymbol{D}(\cdot, \cdot)$ be a distribution divergence. For any fixed prompt $\boldsymbol{c}$, the training objective is defined as:

$$\theta^* = \underset{\theta}{\arg\max} \, \mathbb{E}_{\boldsymbol{x}_0 \sim p_\theta(\boldsymbol{x}_0|\boldsymbol{c})} \big[ r(\boldsymbol{x}_0, \boldsymbol{c}) \big], \quad \text{s.t. } \boldsymbol{D}(p_\theta(\cdot|\boldsymbol{c}), p_{ref}(\cdot|\boldsymbol{c})) \le \delta \tag{3.2}$$

The constraint $\boldsymbol{D}(p_\theta(\cdot|\boldsymbol{c}), p_{ref}(\cdot|\boldsymbol{c})) \le \delta$ is often referred to as a *trust-region* in literature of reinforcement learning (Schulman, 2015; Schulman et al., 2017). This objective (3.2) is equivalent to the minimization of objective (3.3) that adds the constraining term onto the negative reward:

$$\theta^* = \underset{\theta}{\arg\min} \, \mathbb{E}_{\boldsymbol{x}_0 \sim p_\theta(\boldsymbol{x}_0|\boldsymbol{c})} \big[ -\alpha r(\boldsymbol{x}_0, \boldsymbol{c}) \big] + \boldsymbol{D}(p_\theta, p_{ref}) \tag{3.3}$$

Here $\alpha$ is a coefficient that balances reward influences and $\boldsymbol{D}(\cdot)$ acts as a regularization term. Traditional reinforcement learning from human feedback (RLHF) methods in large language models (Ouyang et al., 2022) use the Kullback-Leibler divergences for regularization. Recent work (Luo, 2024) has studied using the integral of KL divergence in objective (3.3) to train one-step text-to-image generators. However, since the KL divergences are defined with the density ratio of two distributions, any misalignment of density supports will lead to severe numerical instability, potentially resulting in annoying mode-seeking behavior (Bishop, 2006). Besides, some recent works have shown that score-based divergences (Zhou et al., 2024b; Luo et al., 2024c) result in better and more stable performance than KL divergences (Luo et al., 2024b; Yin et al., 2023; Nguyen & Tran, 2023) in the literature of diffusion distillation. Motivated by the above two inspirations, we propose a novel score-based online PPO algorithm that uses a general family of score-based divergence as regularization in this paper. We provide solid theoretical foundations of score-based regularization and its better empirical performances through large-scale text-to-image models.

### 3.1 General Score-based Divergences

Different from KL divergence, we can define the regularization term $\boldsymbol{D}(p_\theta, p_{ref})$ via the following general score-based divergence. Assume $\mathbf{d} : \mathbb{R}^d \to \mathbb{R}$ is a scalar-valued proper distance function (i.e., a non-negative function that satisfies $\forall \boldsymbol{x}, \mathbf{d}(\boldsymbol{x}) \ge 0$ and $\mathbf{d}(\boldsymbol{x}) = 0$ if and only if $\boldsymbol{x} = \mathbf{0}$). Given a parameter-independent sampling distribution $\pi_t$ that has large distribution support, we can formally define a time-integral score divergence as

$$\mathbf{D}^{[0,T]}(p_\theta, p_{ref}) := \int_{t=0}^{T} w(t) \mathbb{E}_{\boldsymbol{x}_t \sim \pi_t} \Big\{ \mathbf{d}(\boldsymbol{s}_{p_{\theta,t}}(\boldsymbol{x}_t) - \boldsymbol{s}_{q_t}(\boldsymbol{x}_t)) \Big\} \mathrm{d}t, \tag{3.4}$$

where $p_{\theta,t}$ and $q_t$ denote the marginal densities of the diffusion process (2.1) at time $t$ initialized with $p_{\theta,0} = p_\theta$ and $q_0 = p_{ref}$ respectively. $w(t)$ is an integral weighting function. Clearly, we have $\mathbf{D}^{[0,T]}(p_\theta, p_{ref}) = 0$ if and only if $p_\theta(\boldsymbol{x}_0) = p_{ref}(\boldsymbol{x}_0)$, $a.s.$ $\pi_0$.

---

**Algorithm 1:** Diff-Instruct* for training human-preferred one-step text-to-image generators.

---

**Input:** prompt dataset $\mathcal{C}$, generator $g_\theta(\boldsymbol{x}_0|\boldsymbol{z}, \boldsymbol{c})$, prior distribution $p_z$, reward model $r(\boldsymbol{x}, \boldsymbol{c})$,
reward model scale $\alpha_{rew}$, CFG reward scale $\alpha_{cfg}$, reference diffusion model
$\boldsymbol{s}_{ref}(\boldsymbol{x}_t|c, \boldsymbol{c})$, assistant diffusion $\boldsymbol{s}_\psi(\boldsymbol{x}_t|t, \boldsymbol{c})$, forward diffusion $p_t(\boldsymbol{x}_t|\boldsymbol{x}_0)$ (2.1),
assistant diffusion updates rounds $K_{TA}$, time distribution $\pi(t)$, diffusion model
weighting $\lambda(t)$, generator loss time weighting $w(t)$.

**while** *not converge* **do**

    freeze $\theta$, update $\psi$ for $K_{TA}$ rounds using SGD by minimizing

$$\mathcal{L}(\psi) = \mathbb{E}_{\substack{\boldsymbol{c}\sim\mathcal{C}, \boldsymbol{z}\sim p_z, t\sim\pi(t) \\ \boldsymbol{x}_0 = g_\theta(\boldsymbol{z}|\boldsymbol{c}), \boldsymbol{x}_t|\boldsymbol{x}_0\sim p_t(\boldsymbol{x}_t|\boldsymbol{x}_0)}} \lambda(t)\|\boldsymbol{s}_\psi(\boldsymbol{x}_t|t, \boldsymbol{c}) - \nabla_{\boldsymbol{x}_t}\log p_t(\boldsymbol{x}_t|\boldsymbol{x}_0)\|_2^2 \mathrm{d}t.$$

    freeze $\psi$, update $\theta$ using SGD by minimizing loss

$$\mathcal{L}_{DI*}(\theta) = \mathbb{E}_{\substack{\boldsymbol{c}\sim\mathcal{C}, \boldsymbol{z}\sim p_z, \\ \boldsymbol{x}_0 = g_\theta(\boldsymbol{z}, \boldsymbol{c})}}\Bigg\{ -\alpha_{rew}\cdot r(\boldsymbol{x}_0, \boldsymbol{c}) + \mathbb{E}_{\substack{t\sim\pi(t), \\ \boldsymbol{x}_t|\boldsymbol{x}_0\sim p_t(\boldsymbol{x}_t|\boldsymbol{x}_0)}}\Bigg[$$
$$- w(t)\big\{\mathbf{d}'(\boldsymbol{s}_\psi(\boldsymbol{x}_t|t, \boldsymbol{c}) - \boldsymbol{s}_{ref}(\boldsymbol{x}_t|t, \boldsymbol{c}))\big\}^T\big\{\boldsymbol{s}_\psi(\boldsymbol{x}_t|t, \boldsymbol{c}) - \nabla_{\boldsymbol{x}_t}\log p_t(\boldsymbol{x}_t|\boldsymbol{x}_0)\big\}$$
$$+ \alpha_{cfg}\cdot w(t)\big\{\boldsymbol{s}_{ref}(\text{sg}[\boldsymbol{x}_t]|t, \boldsymbol{c}) - \boldsymbol{s}_{ref}(\text{sg}[\boldsymbol{x}_t]|t, \varnothing)\big\}^T\boldsymbol{x}_t\Bigg]\Bigg\} \quad (3.7)$$

**end**
**return** $\theta, \psi$.

---

## 3.2 DIFF-INSTRUCT*

Recall that $g_\theta$ is a one-step model, therefore samples from $p_\theta$ can be implemented through a direct mapping $\boldsymbol{x}_0 = g_\theta(\boldsymbol{z}|\boldsymbol{c})$. With the score-based regularization term (3.4), for each given text prompt $\boldsymbol{c}$, we can formally write down our training objective to minimize as:

$$\mathcal{L}_{Orig}(\theta) = \mathbb{E}_{\substack{\boldsymbol{z}\sim p_z, \\ \boldsymbol{x}_0 = g_\theta(\boldsymbol{z}, \boldsymbol{c})}}\big[ -\alpha r(\boldsymbol{x}_0, \boldsymbol{c})\big] + \mathbf{D}^{[0,T]}(p_\theta, p_{ref}) \quad (3.5)$$

Now we are ready to reveal the objective of Diff-Instruct* that we use to train human-preferred one-step generator $g_\theta$. Notice that directly minimizing objective (3.5) is intractable because we do not know the relationship between $\theta$ and corresponding $p_{\theta,t}$. However, we show in Theorem 3.1 that an equivalent tractable loss (3.6) will have the same $\theta$ gradient as the intractable loss function (3.5):

$$\mathcal{L}_{DI*}(\theta) = \mathbb{E}_{\substack{\boldsymbol{z}\sim p_z, \\ \boldsymbol{x}_0 = g_\theta(\boldsymbol{z})}}\Bigg[ -\alpha r(\boldsymbol{x}_0, \boldsymbol{c}) \quad\quad\quad\quad\quad\quad\quad\quad\quad (3.6)$$
$$+ \int_{t=0}^{T} w(t)\mathbb{E}_{\substack{\boldsymbol{x}_t|\boldsymbol{x}_0 \\ \sim p_t(\boldsymbol{x}_t|\boldsymbol{x}_0)}}\Big\{ -\mathbf{d}'(\boldsymbol{y}_t)\Big\}^T\Big\{\boldsymbol{s}_{p_{\text{sg}[\theta],t}}(\boldsymbol{x}_t) - \nabla_{\boldsymbol{x}_t}\log p_t(\boldsymbol{x}_t|\boldsymbol{x}_0)\Big\}\mathrm{d}t\Bigg]$$

with $\boldsymbol{y}_t := \boldsymbol{s}_{p_{\text{sg}[\theta],t}}(\boldsymbol{x}_t) - \boldsymbol{s}_{q_t}(\boldsymbol{x}_t)$.

**Theorem 3.1.** Under mild assumptions, if we take the sampling distribution in (3.4) as $\pi_t = p_{\text{sg}[\theta],t}$, then the $\theta$ gradient of (3.5) is the same as the objective (3.6): $\frac{\partial}{\partial\theta}\mathcal{L}_{Orig}(\theta) = \frac{\partial}{\partial\theta}\mathcal{L}_{DI*}(\theta)$.

We will give the proof in Appendix A.1. In practice, we can use another assistant diffusion model $\boldsymbol{s}_\psi(\boldsymbol{x}_t, t)$ to approximate the generator model's score function $\boldsymbol{s}_{p_{\text{sg}[\theta],t}}(\boldsymbol{x}_t)$ pointwise, which was also done in the literature of diffusion distillations works such as Zhou et al. (2024a;b); Luo et al. (2024b;c); Yin et al. (2023; 2024). Therefore, we can alternate between 1) updating the assistant diffusion $\boldsymbol{s}_\psi(\boldsymbol{x}_t, t)$ using generator-generated samples (which are efficient) and 2) updating the generator by minimizing the tractable objective (3.6). We name our training method that minimizes the objective $\mathcal{L}_{DI*}(\theta)$ in (3.6) the Diff-Instruct* because it is inspired by Diff-Instruct(Luo et al., 2024b) and Diff-Instruct++(Luo, 2024) that involves an additional diffusion model and a reward model to train one-step generators.

## 3.3 DECOUPLING THE EXPLICIT REWARD AND IMPLICIT GUIDANCE-BASED REWARDS

**Classifier-free Guidance Corresponds to Implicit Reward.** In previous sections, we have shown in theory that with explicitly available reward models, we can readily train the one-step generator to

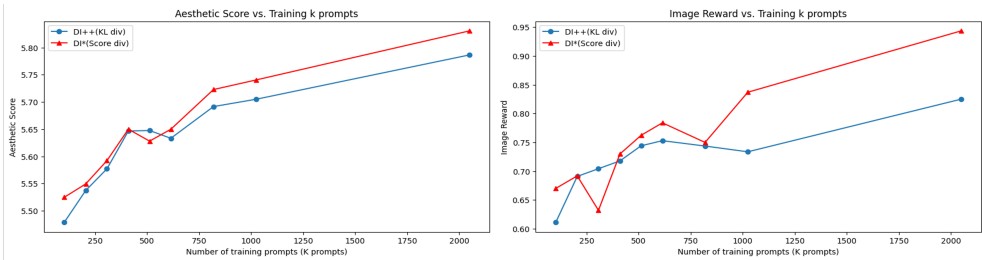

Figure 3: Comparison of Aesthetic Scores and Image Reward on 1K MSCOCO 2017 validation prompts of Score-based (DI*) and KL divergence (DI++(Luo, 2024)) for alignment with (3.5).

align with human preference. In this section, we enhance the DI* by incorporating the classifier-free reward that is implied by the classifier-free guidance of diffusion models.

The classifier-free guidance (Ho & Salimans, 2022) (CFG) uses a modified score function of a form

$$\widetilde{\boldsymbol{s}}_{ref}(\boldsymbol{x}_t, t|\boldsymbol{c}) \coloneqq \boldsymbol{s}_{ref}(\boldsymbol{x}_t, t|\boldsymbol{\varnothing}) + \omega\big\{\boldsymbol{s}_{ref}(\boldsymbol{x}_t, t|\boldsymbol{c}) - \boldsymbol{s}_{ref}(\boldsymbol{x}_t, t|\boldsymbol{\varnothing})\big\}$$

to replace the original conditions score function $\boldsymbol{s}_{ref}(\boldsymbol{x}_t, t|\boldsymbol{c})$. Using CFG for diffusion models empirically leads to better sampling quality.

As is first pointed out by Luo (2024), the classifier-free guidance is related to an implicit reward function. In this part, we derive a tractable loss function that minimizes the so-called classifier-free reward, which we use together with the explicit reward $r(\cdot, \cdot)$ in DI*.

**Theorem 3.2.** Under mild conditions, if we set an implicit reward function as (3.9), the loss (3.8)

$$\mathcal{L}_{cfg}(\theta) = \int_{t=0}^{T} \mathbb{E}_{\substack{\boldsymbol{z} \sim p_z, \boldsymbol{x}_0 = g_\theta(\boldsymbol{z}, \boldsymbol{c}) \\ \boldsymbol{x}_t|\boldsymbol{x}_0 \sim p(\boldsymbol{x}_t|\boldsymbol{x}_0)}} w(t)\bigg\{\boldsymbol{s}_{ref}(\mathrm{sg}[\boldsymbol{x}_t]|t, \boldsymbol{c}) - \boldsymbol{s}_{ref}(\mathrm{sg}[\boldsymbol{x}_t]|t, \boldsymbol{\varnothing})\bigg\}^{T} \boldsymbol{x}_t \mathrm{d}t \qquad (3.8)$$

has the same gradient as the negative implicit reward function (3.9)

$$-r(\boldsymbol{x}_0, \boldsymbol{c}) = -\int_{t=0}^{T} \mathbb{E}_{\boldsymbol{x}_t \sim p_{\theta,t}} w(t) \log \frac{p_{ref}(\boldsymbol{x}_t|t, \boldsymbol{c})}{p_{ref}(\boldsymbol{x}_t|t)} \mathrm{d}t. \qquad (3.9)$$

The notation $\mathrm{sg}[\boldsymbol{x}_t]$ means detaching the $\theta$ gradient on $\boldsymbol{x}_t$. We give the proof of Theorem 3.2 in Appendix A.2. Theorem 3.2 gives a tractable loss function (3.8) aiming to minimize the negative classifier-free reward function. Therefore, we can scale and add this loss $\mathcal{L}_{cfg}(\theta)$ (3.8) to the DI* loss (3.6) to balance the effects of explicit reward and implicit CFG reward.

### 3.4 THE PRACTICAL ALGORITHM

Now it is time for us to introduce the practical algorithm. As Algorithm 1 (and a more executable version in Algorithm 2) shows, the DI* involves three models, with one generator model $g_\theta$, one reference diffusion model $\boldsymbol{s}_{ref}$ and one assistant diffusion model $\boldsymbol{s}_\psi$. The reference diffusion does not need to be trained, while the generator and the assistant diffusion are updated alternatively. Two hyper-parameters, the $\alpha_{rew}$ and $\alpha_{cfg}$ control the strength of the explicit reward and the implicit CFG reward during training. The explicit reward model can either be an off-the-shelf reward model, such as the CLIP similarity score (Radford et al., 2021), or the Image Reward (Xu et al., 2023a) or trained in-house with researchers' internal human feedback data. Due to page limitations, we put a discussion about the meanings of hyper-parameters in Appendix B.2.

**Flexibility in Distance Functions.** Clearly, various choices of distance function $\mathbf{d}(.)$ result in different training algorithms. For instance, $\mathbf{d}(\boldsymbol{y}_t) = \|\boldsymbol{y}_t\|_2^2$ is a naive choice. Interestingly, such a distance function has been studied in pure diffusion distillation literature in Zhou et al. (2024b;a); Luo et al. (2024c). In this paper, we draw inspiration from (Luo et al., 2024c) and find that using the so-called pseudo-Huber distance leads to better performance. The distance and corresponding loss writes $\boldsymbol{d}(\boldsymbol{y}) \coloneqq \sqrt{\|\boldsymbol{y}_t\|_2^2 + c^2} - c$, and

$$\mathbf{D}^{[0,T]}(p_\theta, p_{ref}) = -\bigg\{\frac{\boldsymbol{y}_t}{\sqrt{\|\boldsymbol{y}_t\|_2^2 + c^2}}\bigg\}^{T}\bigg\{\boldsymbol{s}_\psi(\boldsymbol{x}_t, t) - \nabla_{\boldsymbol{x}_t} \log p_t(\boldsymbol{x}_t|\boldsymbol{x}_0)\bigg\}. \qquad (3.10)$$

Here $\boldsymbol{y}_t \coloneqq \boldsymbol{s}_{p_{\mathrm{sg}[\theta],t}}(\boldsymbol{x}_t) - \boldsymbol{s}_{q_t}(\boldsymbol{x}_t)$.

## 4 RELATED WORKS

**Diffusion Distillation Through Divergence Minimization.** Diff-Instruct* is inspired by research on diffusion distillation (Luo, 2023) which aims to minimize certain distribution divergence to train one-step generators. Luo et al. (2024b) first study the diffusion distillation by minimizing the Integral KL divergence. Yin et al. (2023) generalize such a concept and add a data regression loss for better performance. Zhou et al. (2024b) study the distillation by minimizing the Fisher divergence. Luo et al. (2024c) study the distillation using the general score-based divergence. Many other works also introduced additional techniques and improved the performance (Geng et al., 2023; Kim et al., 2023; Song et al., 2023; Song & Dhariwal; Nguyen & Tran, 2023; Song et al., 2024; Yin et al., 2024; Zhou et al., 2024a; Heek et al., 2024; Xie et al., 2024; Salimans et al., 2024; Geng et al., 2024).

**Preference Alignment for Diffusion Models and One-step Generators.** In recent years, many works have emerged trying to align diffusion models with human preferences. There are three main lines of alignment methods for diffusion models. 1) The first kind of method fine-tunes the diffusion model over a specifically curated image-prompt dataset (Dai et al., 2023; Podell et al., 2023). 2) the second line of methods tries to maximize some reward functions either through the multi-step diffusion generation output (Prabhudesai et al., 2023; Clark et al., 2023; Lee et al., 2023) or through policy gradient-based RL approaches (Fan et al., 2024; Black et al., 2023). For these methods, the backpropagation through the multi-step diffusion generation output is expensive and hard to scale. 3) the third line, such as Diffusion-DPO (Wallace et al., 2024), Diffusion-KTO (Yang et al., 2024), tries to directly improve the diffusion model's human preference property with raw collected data instead of reward functions. Besides the human preference alignment of diffusion models, Diff-Instruct++(Luo, 2024) recently arose as the first attempt to improve human preferences for one-step generators. Though inspired by DI++, DI* uses score-based divergences which are technically different from the KL divergence used in DI++. Besides, as we show in Section 5.2, DI* archives better performances than DI++ in Luo (2024).

## 5 EXPERIMENTS

### 5.1 EXPERIMENT SETTTINGS

**Experiment Settings for SD1.5 and SDXL Experiment.** For experiments of SD1.5, we use the open-sourced SD1.5 of a resolution of $512 \times 512$ as our reference diffusion in Algorithm 1. We implement our experiments based on SiD-LSG (Zhou et al., 2024a) codebase. We construct the one-step generator with the same architecture as the reference SD1.5 model, following the same configuration of SiD-LSG. We use the prompts of the LAION-AESTHETIC dataset with an aesthetic score larger than 6.25, which resulting a total of 3M text prompts. To better explore the advantages of our score-based divergence over KL divergence, we refer to a recent work (Luo, 2024) that uses KL divergence for training and conducts a detailed comparison between two divergences. We explore different combinations and find that ($\alpha_{rew} = 1000, \alpha_{cfg} = 1.5$) in Algorithm 1 is the best. For SD1.5 experiments, we find that training one-step generators from scratch leads to longer training, therefore we initialize our generator with the weights of the SiD-LSG pre-trained one-step model. For experiments of SDXL, we use the SDXL(Podell et al., 2023) as the teacher model and its architecture as the one-step student model. We initialize the one-step generator with the pre-trained DMD2-SDXL-1step model (Yin et al., 2024), which is a pretty good initialization based on SDXL architectures. We follow similar settings as the SD1.5 experiment: using a scale for explicit ImageReward of 1000 and a scale for implicit CFG reward of 8.0.

**Experiment Settings for PixelArt-$\alpha$ Experiment.** PixArt-$\alpha$ is a high-quality DiT-based(Peebles & Xie, 2022) open-sourced text-to-image diffusion model. We use the DiT architecture and the 0.6B PixArt-$\alpha$ of a resolution of $512 \times 512$ as our reference diffusion to demonstrate the compatibility of DI* for different neural network architectures. We use the prompts from the training dataset of PixArt-$\alpha$ (the SAM-Recaptioned Dataset), resulting in a total of 10M prompts. After exploring different combinations, we find that ($\alpha_{rew} = 10, \alpha_{cfg} = 4.5$) in Algorithm 1 gives the best performances. For DiT generators with a total of 0.6B parameters, we find that training from scratch without initialization results in strong results. Therefore we do not apply special initialization for DiT-based one-step generators. Our best DiT-DI* model in Table 2 is trained from scratch with 8 H100-80G GPUs for 200000 iterations with a batch size of 128. The total wall-clock training costs

Table 1: Quantitative comparisons of text-to-image models on **MSCOCO-2017 validation** prompts (the upper part) and **Parti(Yu et al., 2022)** Prompts (the under part). DI* is short for Diff-Instruct*. $\alpha_r$ and $\alpha_c$ are short for $\alpha_{rew}$ and $\alpha_{cfg}$ in Algorithm 1. † means our implementation. Data means the model needs image data for training. Sampling means the model needs to draw samples from reference diffusion models. Reward means the model needs a human reward model for training. † indicates our implementation. ‡ indicates the same 4-step model of DMD2 but with different inference steps.

| MODEL | STEPS | TYPE | PARAMS | IMAGE REWARD | AES SCORE | PICK SCORE | CLIP SCORE | ADDITIONAL REQUIREMENTS |
|---|---|---|---|---|---|---|---|---|
| SD15-DPO(WALLACE ET AL., 2024) | 15 | UNET | 0.86B | 0.20 | 5.29 | 0.214 | 31.07 | DATA, SAMPLING |
| SD15-DPO(WALLACE ET AL., 2024) | 25 | UNET | 0.86B | 0.28 | 5.37 | 0.218 | 31.25 | PREFERENCE DATA |
| SD15-LCM(LUO ET AL., 2023A) | 1 | UNET | 0.86B | -1.58 | 5.04 | 0.194 | 27.20 | DATA, SAMPLING |
| SD15-LCM(LUO ET AL., 2023A) | 4 | UNET | 0.86B | -0.23 | 5.40 | 0.214 | 30.11 | DATA, SAMPLING |
| SD15-TCD(ZHENG ET AL., 2024) | 1 | UNET | 0.86B | -1.49 | 5.10 | 0.196 | 28.30 | DATA, SAMPLING |
| SD15-TCD(ZHENG ET AL., 2024) | 4 | UNET | 0.86B | -0.04 | 5.28 | 0.212 | 30.43 | DATA, SAMPLING |
| PERFLOW(YAN ET AL., 2024) | 4 | UNET | 0.86B | -0.20 | 5.51 | 0.211 | 29.54 | DATA, SAMPLING |
| SD15-HYPER(REN ET AL., 2024) | 1 | UNET | 0.86B | 0.28 | 5.49 | 0.214 | 30.82 | DATA, SAMPLING |
| SD15-HYPER(REN ET AL., 2024) | 4 | UNET | 0.86B | 0.42 | 5.41 | 0.217 | 31.03 | REWARD, SEG-MODEL |
| SD15-INSTAFLOW(LIU ET AL., 2023) | 1 | UNET | 0.86B | -0.16 | 5.03 | 0.207 | 30.68 | DATA, SAMPLING |
| SDXL-BASE(ROMBACH ET AL., 2022) | 25 | UNET | 2.6B | 0.74 | 5.57 | 0.226 | 31.83 | IMAGE-TEXT |
| SDXL-BASE(ROMBACH ET AL., 2022) | 15 | UNET | 2.6B | 0.68 | 5.56 | 0.224 | 31.99 | IMAGE-TEXT |
| SDXL-DMD2‡-1024(YIN ET AL., 2024) | 1 | UNET | 2.6B | 0.82 | 5.45 | 0.224 | 31.78 | IMAGE-TEXT |
| SDXL-DMD2‡-1024(YIN ET AL., 2024) | 4 | UNET | 2.6B | 0.87 | 5.52 | **0.231** | 31.50 | IMAGE-TEXT |
| SDXL-DMD2‡-512(YIN ET AL., 2024) | 1 | UNET | 2.6B | 0.36 | 5.03 | 0.215 | 31.54 | IMAGE-TEXT |
| SDXL-DMD2‡-512(YIN ET AL., 2024) | 4 | UNET | 2.6B | -0.18 | 5.17 | 0.206 | 29.28 | IMAGE-TEXT |
| SD15-DMD2-512(YIN ET AL., 2024) | 1 | UNET | 2.6B | -0.12 | 5.24 | 0.211 | 30.00 | IMAGE-TEXT |
| SD21-TURBO(SAUER ET AL., 2023B) | 1 | UNET | 0.86B | 0.56 | 5.47 | 0.225 | 31.50 | IMAGE-TEXT |
| SD15-BASE(ROMBACH ET AL., 2022) | 15 | UNET | 0.86B | 0.08 | 5.25 | 0.212 | 30.99 | IMAGE-TEXT |
| SD15-BASE(ROMBACH ET AL., 2022) | 25 | UNET | 0.86B | 0.22 | 5.32 | 0.216 | 31.13 | IMAGE-TEXT |
| PIXELART-$\alpha$-512(CHEN ET AL., 2023) | 25 | DIT | 0.6B | 0.82 | 6.01 | 0.227 | 31.20 | IMAGE-TEXT |
| PIXELART-$\alpha$-512(CHEN ET AL., 2023) | 15 | DIT | 0.6B | 0.82 | 6.03 | 0.226 | 31.16 | IMAGE-TEXT |
| SD15-SIDLSG(ZHOU ET AL., 2024A) | 1 | UNET | 0.86B | -0.18 | 5.16 | 0.210 | 30.04 | TEXT |
| SDXL-DMD2-1024(YIN ET AL., 2024) | 1 | UNET | 2.6B | 0.85 | 5.46 | 0.225 | 31.86 | IMAGE-TEXT |
| SD15-DI++(LUO, 2024)† | 1 | UNET | 0.86B | 0.82 | 5.78 | 0.219 | 30.30 | REWARD |
| DIT-DI++(LUO, 2024)† | 1 | DIT | 0.6B | 1.24 | 6.19 | 0.225 | 30.80 | REWARD |
| **SD15-DI***($\alpha_r = 0, \alpha_c = 1.5$) | 1 | UNET | 0.86B | 0.34 | 5.27 | 0.217 | 30.83 | REWARD |
| **SD15-DI***($\alpha_r = 100, \alpha_c = 1.5$) | 1 | UNET | 0.86B | 0.62 | 5.44 | 0.218 | 30.76 | REWARD |
| **SD15-DI***($\alpha_r = 1000, \alpha_c = 4.5$) | 1 | UNET | 0.86B | 0.73 | 5.56 | 0.219 | 30.71 | REWARD |
| **SD15-DI***($\alpha_r = 1000, \alpha_c = 1.5$) | 1 | UNET | 0.86B | 0.94 | 5.83 | 0.220 | 30.49 | REWARD |
| **SDXL-DI*-1024**($\alpha_r = 1000, \alpha_c = 8.0$) | 1 | UNET | 2.6B | 0.88 | 5.56 | 0.225 | **32.07** | REWARD |
| **DIT-DI***($\alpha_r = 1, \alpha_c = 4.5$) | 1 | DIT | 0.6B | 0.98 | 6.02 | 0.225 | 31.00 | REWARD |
| **DIT-DI***($\alpha_r = 10, \alpha_c = 4.5$) | 1 | DIT | 0.6B | **1.31** | **6.30** | 0.225 | 30.84 | REWARD |
| SDXL-BASE(ROMBACH ET AL., 2022) | 15 | UNET | 2.6B | 0.69 | 5.68 | 0.224 | 32.76 | IMAGE-TEXT |
| PIXELART-$\alpha$-512(CHEN ET AL., 2023) | 15 | DIT | 0.6B | 0.96 | 6.00 | **0.227** | 31.76 | IMAGE-TEXT |
| SDXL-DMD2-1024(YIN ET AL., 2024) | 1 | UNET | 2.6B | 0.94 | 5.53 | 0.225 | 33.00 | IMAGE-TEXT |
| **SDXL-DI*-1024**($\alpha_r = 1000, \alpha_c = 8.0$) | 1 | UNET | 2.6B | 1.06 | 5.61 | 0.225 | **33.27** | REWARD |
| **DIT-DI***($\alpha_r = 1, \alpha_c = 4.5$) | 1 | DIT | 0.6B | 0.96 | 6.06 | 0.224 | 31.11 | REWARD |
| **DIT-DI***($\alpha_r = 10, \alpha_c = 4.5$) | 1 | DIT | 0.6B | **1.26** | **6.23** | 0.225 | 30.64 | REWARD |

is less than 30 hours. As a comparison, other industry models in Table 1 usually require hundreds of A100 GPU days for training. Such a low cost might benefit from the property that DI* needs neither real image data nor samples from reference diffusion models.

**Quantitative Evaluations Metrics.** We compare our generators with other leading open-sourced models that are either based on SD1.5 diffusion models or larger models such as SDXL. For all models, we compute four standard scores in Table 1: the Image Reward (Xu et al., 2023a), the Aesthetic Score (Schuhmann, 2022), the PickScore(Kirstain et al., 2023), and the CLIP score(Radford et al., 2021) on the same 1K prompts randomly sampled from COCO-2017-validation(Lin et al., 2014) set on the same computing devices. We have compared the results on 1K prompts and 30K COCO prompts and found similar results. Since our training prompts do not involve COCO validation prompts, the evaluation of COCO prompts can be viewed as an out-of-training set evaluation. We also evaluate models with Human Preference Score v2.0 (HPSv2.0) (Wu et al., 2023) in Table 2. The HPSv2.0 is a standard score for evaluating models' human preferences across different styles.

## 5.2 Performances and Findings

**Quantitative Comparison: DI\* Achieves SoTA Human Preference Scores.** As Table 1 shows, our best 0.6B DiT-DI\* model outperforms all other open-sourced models. It achieves a COCO (out-of-sample) Image Reward of **1.31**, which is **50%** better than the second best 2.6B SDXL-DMD2-4Step model of 1024 resolution. It also outperforms the SDXL model with a margin of **70%**. It also shows an Aesthetic Score of **6.30**, which is **14.3%** better than the 2.6B SDXL-DMD2-4Step model. Among SD1.5-based models, our best SD1.5-DI\* model outperforms other models with significant margins. This result demonstrates the compatibility of DI\* across UNet and DiT architectures.

As Table 2 shows, our 0.6B DiT-DI\* one-step model achieves a record-breaking HPSv2.0 score of **28.70** across open-sourced text-to-image models. It clearly outperforms the autoregressive models such as 6B CogView2 (Ding et al., 2022), diffusion models such as 5.5B Dalle-E 2 (Ramesh et al., 2022), 5B GLIDE(Nichol et al., 2021), 2.6B SDXL(Podell et al., 2023), and 4.3B DeepFloyd-XL with up to 30+ generation steps. Such a lightweight, high-performance, and high-efficiency model will bring big impacts on applications that need real-time generations.

**Qualitative Comparisons.** As Figure 1 and Figure 2 show, models trained with DI\* show better layouts, richer colors, and more aesthetic face preferences. We find such an advantage is stronger for scene generations. Figure 4 shows a visualization of the SD1.5-DI\* model in Table 2 before and after alignments with DI\*. It shows that models after alignment produce images with richer colors. We also evaluate the COCO-FID values of SD1.5-based one-step models with and without preference alignment with DI\*. The FID of the alignment model is 18.44, while the un-alignment model (the SiD-LSG which we use as the initialization of the one-step generator) has an FID of 8.27. This phenomenon suggests that only using ImageReward(which tends to be subjective) to align models with DI\* can potentially harm objective metrics such as FID and CLIPScore. A naive solution to hack CLIPScore is to simply include the CLIP score and PickScore as two other sources of reward when using DI\*, which will definitely improve these scores. However, since we want to verify the generalization ability of one reward to others using DI\*, we only use the ImageReward and observed other scores in this paper.

**Score-based Divergence is Better than KL Divergence.** In Figure 3 we compare the use of score-based divergence of DI\* and the KL divergence of DI++(Luo, 2024) for RLHF regularization in (3.5). We fix the $\alpha_{rew} = 1000$ and $\alpha_{cfg} = 1.5$ for both DI++ and DI\* and use the same Image Reward as an explicit reward model for training the SD1.5-based one-step generators. As we can see in Figure 3, for each iteration, DI\* has a better score than DI++. Besides, DI\* achieves the best final results. The reason for the worse performance of KL divergence might be its definition, which involves the ratio of two distributions, which may lead to unstable numerical performances when two distributions have misaligned density supports.

## 6 Conclusion and Limitations

In this paper, we present Diff-Instruct\*, a novel approach for aligning human-preferred one-step text-to-image generators. By formulating the training objective as a maximization of expected human reward functions with a score-based divergence regularization, we have developed practical losses and easy-to-implement algorithms. Our results show that DiT-based one-step text-to-image generators trained with DI\* achieve new state-of-the-art human preference performances.

Nonetheless, DI\* has its limitations. We empirically find three typical mistakes the generator often makes. (1) The Generator Sometimes Generates Bad Human Faces and Hands. (2) The aligned model still can not count correctly. Figure 5 shows some occasional bad generation cases. We believe that consistently improving both the generator architecture and the reward models will lead to better models. Besides, we are also interested in following directions that call for further research. First, in this paper, we only study using only one reward model to train the generator. However, training generators with multiple rewards is still unexplored. Second, the DI\* needs a pre-trained reward model. However, methods like DPO (Rafailov et al., 2024; Wallace et al., 2024) have put a new setup that trains generative models directly using human feedback data. Whether it is possible to develop DPO-like algorithms based on DI\* is a promising research direction. Third, it is also interesting to explore training multistep models instead of one-step generators for better performance. We hope these future directions could contribute more to the community.

Table 2: **HPSv2.0** (upper table) and **HPSv2.1** (under table). We compare open-sourced models regardless of their base model and architecture. † indicates our implementation. ‡ indicates the same 4-step model of DMD2 but with different inference steps.

| MODEL | ANIMATION | CONCEPT-ART | PAINTING | PHOTO | AVERAGE |
|---|---|---|---|---|---|
| GLIDE (NICHOL ET AL., 2021) | 23.34 | 23.08 | 23.27 | 24.50 | 23.55 |
| LAFITE (ZHOU ET AL., 2022) | 24.63 | 24.38 | 24.43 | 25.81 | 24.81 |
| VQ-DIFFUSION (GU ET AL., 2022) | 24.97 | 24.70 | 25.01 | 25.71 | 25.10 |
| FUSEDREAM (LIU ET AL., 2021) | 25.26 | 25.15 | 25.13 | 25.57 | 25.28 |
| LATENT DIFFUSION (ROMBACH ET AL., 2022) | 25.73 | 25.15 | 25.25 | 26.97 | 25.78 |
| COGVIEW2 (DING ET AL., 2022) | 26.50 | 26.59 | 26.33 | 26.44 | 26.47 |
| DALL·E MINI | 26.10 | 25.56 | 25.56 | 26.12 | 25.83 |
| VERSATILE DIFFUSION (XU ET AL., 2023B) | 26.59 | 26.28 | 26.43 | 27.05 | 26.59 |
| VQGAN + CLIP (ESSER ET AL., 2021) | 26.44 | 26.53 | 26.47 | 26.12 | 26.39 |
| DALL·E 2 (RAMESH ET AL., 2022) | 27.34 | 26.54 | 26.68 | 27.24 | 26.95 |
| STABLE DIFFUSION V1.4 (ROMBACH ET AL., 2022) | 27.26 | 26.61 | 26.66 | 27.27 | 26.95 |
| STABLE DIFFUSION V2.0 (ROMBACH ET AL., 2022) | 27.48 | 26.89 | 26.86 | 27.46 | 27.17 |
| EPIC DIFFUSION | 27.57 | 26.96 | 27.03 | 27.49 | 27.26 |
| DEEPFLOYD-XL | 27.64 | 26.83 | 26.86 | 27.75 | 27.27 |
| OPENJOURNEY | 27.85 | 27.18 | 27.25 | 27.53 | 27.45 |
| MAJICMIX REALISTIC | 27.88 | 27.19 | 27.22 | 27.64 | 27.48 |
| CHILLOUTMIX | 27.92 | 27.29 | 27.32 | 27.61 | 27.54 |
| DELIBERATE | 28.13 | 27.46 | 27.45 | 27.62 | 27.67 |
| REALISTIC VISION | 28.22 | 27.53 | 27.56 | 27.75 | 27.77 |
| SDXL-BASE(PODELL ET AL., 2023) | 28.42 | 27.63 | 27.60 | 27.29 | 27.73 |
| SDXL-REFINER(PODELL ET AL., 2023) | 28.45 | 27.66 | 27.67 | 27.46 | 27.80 |
| DREAMLIKE PHOTOREAL 2.0 | 28.24 | 27.60 | 27.59 | 27.99 | 27.86 |
| SD15-DPO-15STEP(WALLACE ET AL., 2024) | 27.11 | 26.75 | 26.70 | 27.30 | 26.97 |
| SD15-DPO-25STEP(WALLACE ET AL., 2024) | 27.54 | 26.97 | 26.99 | 27.49 | 27.25 |
| SD15-LCM-1STEP(LUO ET AL., 2023A) | 23.35 | 23.41 | 23.53 | 23.81 | 23.52 |
| SD15-LCM-4STEP(LUO ET AL., 2023A) | 26.42 | 25.79 | 25.95 | 26.91 | 26.27 |
| SD15-TCD-1STEP(ZHENG ET AL., 2024) | 23.37 | 23.16 | 23.26 | 23.88 | 23.42 |
| SD15-TCD-4STEP(ZHENG ET AL., 2024) | 26.67 | 26.25 | 26.26 | 27.19 | 26.59 |
| SD15-HYPER-1STEP(REN ET AL., 2024) | 27.76 | 27.36 | 27.41 | 27.63 | 27.54 |
| SD15-HYPER-4STEP(REN ET AL., 2024) | 28.04 | 27.39 | 27.42 | 27.89 | 27.69 |
| SD15-INSTAFLOW-1STEP(LIU ET AL., 2023) | 26.07 | 25.80 | 25.89 | 26.32 | 26.02 |
| SD15-PEREFLOW-1STEP(YAN ET AL., 2024) | 25.70 | 25.45 | 25.57 | 25.96 | 25.67 |
| SD15-BOOT-1STEP(GU ET AL., 2023) | 25.29 | 24.40 | 24.61 | 25.16 | 24.86 |
| SD21-SWIFTBRUSH-1STEP(NGUYEN & TRAN, 2023) | 26.91 | 26.32 | 26.37 | 27.21 | 26.70 |
| SD21-TURBO-1STEP(SAUER ET AL., 2023B) | 27.48 | 26.86 | 27.46 | 26.89 | 27.71 |
| SDXL-DMD2‡-1STEP-1024(YIN ET AL., 2024) | 27.67 | 27.02 | 27.01 | 26.94 | 27.16 |
| SDXL-DMD2‡-4STEP-1024(YIN ET AL., 2024) | 28.97 | 27.99 | 27.90 | 28.28 | 28.29 |
| SDXL-DMD2‡-1STEP-512(YIN ET AL., 2024) | 27.70 | 27.07 | 27.02 | 26.94 | 27.18 |
| SDXL-DMD2‡-4STEP-512(YIN ET AL., 2024) | 27.22 | 26.65 | 26.62 | 26.57 | 26.76 |
| SD15-DMD2-1STEP-512(YIN ET AL., 2024) | 26.31 | 25.75 | 25.78 | 26.59 | 26.11 |
| SD15-15STEP(ROMBACH ET AL., 2022) | 26.76 | 26.37 | 26.41 | 27.12 | 26.66 |
| SD15-25STEP(ROMBACH ET AL., 2022) | 27.04 | 26.57 | 26.61 | 27.30 | 26.88 |
| SDXL-BASE-15STEP(PODELL ET AL., 2023) | 28.25 | 27.27 | 27.43 | 27.43 | 27.60 |
| SD15-SIDLSG-1STEP(REPORT)(ZHOU ET AL., 2024A) | 27.39 | 26.65 | 26.58 | 27.30 | 26.98 |
| SD15-SIDLSG-1STEP(ZHOU ET AL., 2024A) | 26.37 | 25.85 | 25.88 | 26.73 | 26.20 |
| PIXELART-$\alpha$-25STEP-512(CHEN ET AL., 2023) | 28.77 | 27.92 | 27.96 | 28.37 | 28.25 |
| PIXELART-$\alpha$-15STEP-512(CHEN ET AL., 2023) | 28.68 | 27.85 | 27.87 | 28.29 | 28.17 |
| SD15-DI++-1STEP(LUO, 2024)† | 28.42 | 27.84 | 28.01 | 28.19 | 28.12 |
| SDXL-DMD2-1STEP-1024(YIN ET AL., 2024)($\alpha_c = 8$) | 28.45 | 27.52 | 27.52 | 27.75 | 27.81 |
| **SD15-DI\*-1STEP**($\alpha_r = 1000, \alpha_c = 1.5$)(OURS) | 28.56 | 28.05 | 28.17 | 28.31 | 28.27 |
| **SDXL-DI\*-1STEP-1024**($\alpha_r = 1000, \alpha_c = 8.0$)(OURS) | 28.74 | 28.04 | 28.14 | 28.12 | 28.26 |
| **DIT-DI\*-1STEP**($\alpha_r = 10, \alpha_c = 4.5$)(OURS) | 28.78 | 28.31 | 28.48 | 28.37 | 28.48 |
| **DIT-DI\*-1STEP**($\alpha_r = 1, \alpha_c = 4.5$)(OURS) | **29.13** | **28.51** | **28.51** | **28.63** | **28.70** |
| SD15-15STEP(ROMBACH ET AL., 2022) | 23.43 | 22.91 | 22.76 | 24.17 | 23.32 |
| SDXL-BASE-15STEP(PODELL ET AL., 2023) | 29.71 | 27.69 | 27.71 | 25.46 | 27.64 |
| PIXLART-$\alpha$-15STEP(CHEN ET AL., 2023) | 31.31 | 29.86 | 29.56 | 28.49 | 29.81 |
| SD15-SIDLSG-1STEP(ZHOU ET AL., 2024A) | 22.54 | 21.53 | 21.37 | 23.09 | 22.13 |
| SDXL-DMD2-1STEP(YIN ET AL., 2024) | 29.72 | 27.96 | 27.64 | 26.55 | 27.97 |
| RCM-IR-2STEP(LI ET AL., 2024) | 29.65 | 31.15 | 32.00 | 31.03 | 30.95 |
| **SD15-DI\*-1STEP**(OURS) | 29.00 | 28.88 | 29.17 | 27.68 | 28.68 |
| **SDXL-DI\*-1STEP-1024**(OURS) | 30.74 | 30.03 | 30.05 | 28.00 | 29.71 |
| **DIT-DI\*-1STEP**(OURS) | **33.05** | **32.37** | **32.10** | **30.07** | **31.90** |

## BROADER IMPACT STATEMENT

This work is motivated by our aim to increase the positive impact of one-step text-to-image generative models toward satisfying human preferences. By default, one-step generators are either trained over large-scale image-caption pair datasets or distilled from pre-trained diffusion models, which convey only subjective knowledge without human instructions.

Our results indicate that the proposed approach is promising for making one-step generative models more aesthetic, and more preferred by human users. In the longer term, alignment failures could lead to more severe consequences, particularly if these models are deployed in safety-critical situations. For instance, if alignment failures occur, the one-step text-to-image model may generate toxic images with misleading information, and horrible images that can potentially be scary to users. We strongly recommend using our human preference alignment techniques together with AI safety checkers for text-to-image generation to prevent undesirable negative impacts.

## REPRODUCIBILITY STATEMENT

We provide extensive details of experimental settings and hyperparameters to reproduce our experimental results. We plan to release our code to ensure transparency and reproducibility of the results.

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

## A  THEORY

### A.1  PROOF OF THEOREM 3.1

*Proof.* Recall that $p_\theta(\cdot)$ is induced by the generator $g_\theta(\cdot)$, therefore the sample is obtained by $\boldsymbol{x}_0 = g_\theta(\boldsymbol{z}|\boldsymbol{c}), \boldsymbol{z} \sim p_z$. The term $\boldsymbol{x}$ contains parameter through $\boldsymbol{x}_0 = g_\theta(\boldsymbol{z}|\boldsymbol{c}), \boldsymbol{z} \sim p_z$. To demonstrate the parameter dependence, we use the notation $p_\theta(\cdot)$. Note that $p_{ref}(\cdot)$ is the reference distribution. The alignment objective writes

$$\mathcal{L}_{Orig}(\theta) = \mathbb{E}_{\substack{\boldsymbol{z} \sim p_z, \\ \boldsymbol{x}_0 = g_\theta(\boldsymbol{z}|\boldsymbol{c})}} \big[ -\alpha r(\boldsymbol{x}_0, \boldsymbol{c}) \big] + \mathbf{D}^{[0,T]}(p_\theta, p_{ref}) \tag{A.1}$$

The first loss term of (A.1) $\alpha r(\boldsymbol{x}_0, \boldsymbol{c})$ is easy to compute by directly pushing the generated sample $\boldsymbol{x}_0$ and the text prompt $\boldsymbol{c}$ into the reward model $r(\cdot, \cdot)$. However, the second loss term (A.2) is intractable because we do not explicitly know the relation between $\theta$ and $p_{\theta,t}(\cdot)$.

$$\mathbf{D}^{[0,T]}(p_\theta, p_{ref}) := \int_{t=0}^{T} w(t) \mathbb{E}_{\boldsymbol{x}_t \sim \pi_t} \Big\{ \mathbf{d}(\boldsymbol{s}_{p_{\theta,t}}(\boldsymbol{x}_t) - \boldsymbol{s}_{q_t}(\boldsymbol{x}_t)) \Big\} \mathrm{d}t, \tag{A.2}$$

We turn to derive the equivalent loss for $\mathbf{D}^{[0,T]}(p_\theta, p_{ref})$. First we take the $\theta$ gradient of (A.2), show

$$\frac{\partial}{\partial \theta} \mathbf{D}^{[0,T]}(p_\theta, p_{ref}) = \frac{\partial}{\partial \theta} \int_{t=0}^{T} w(t) \mathbb{E}_{\boldsymbol{x}_t \sim \pi_t} \Big\{ \mathbf{d}(\boldsymbol{s}_{p_{\theta,t}}(\boldsymbol{x}_t) - \boldsymbol{s}_{q_t}(\boldsymbol{x}_t)) \Big\} \mathrm{d}t \tag{A.3}$$

$$= \mathbb{E}_{t, \boldsymbol{x}_t \sim \pi_t} w(t) \Big\{ \mathbf{d}'(\boldsymbol{y}_t) \Big\}^T \frac{\partial}{\partial \theta} \boldsymbol{s}_{p_{\theta,t}}(\boldsymbol{x}_t) \tag{A.4}$$

Notice that $p_{\theta,t}(\cdot)$ is induced by first generating samples with one-step generator then adding noise with diffusion process (2.1), we do not know the term $\frac{\partial}{\partial \theta} \boldsymbol{s}_{p_{\theta,t}}(\boldsymbol{x}_t)$. Therefore the gradient formula (A.4) is intractable. However, we will show that a tractable loss function can recover the intractable gradient (A.4), and therefore can be used for minimizing (A.2). Our proof is inspired by the theory from Vincent (2011), Zhou et al. (2024b) and Luo et al. (2024c).

We first present a so-called Score-projection identity (Theorem A.1), which has been studied in Zhou et al. (2024b) and Vincent (2011):

**Theorem A.1.** Let $\boldsymbol{u}(\cdot)$ be a $\theta$-free vector-valued function under mild conditions, the identity holds:

$$\mathbb{E}_{\substack{\boldsymbol{x}_0 \sim p_{\theta,0}, \\ \boldsymbol{x}_t | \boldsymbol{x}_0 \sim q_t(\boldsymbol{x}_t | \boldsymbol{x}_0)}} \boldsymbol{u}(\boldsymbol{x}_t)^T \Big\{ \boldsymbol{s}_{p_{\theta,t}}(\boldsymbol{x}_t) - \nabla_{\boldsymbol{x}_t} \log q_t(\boldsymbol{x}_t | \boldsymbol{x}_0) \Big\} = 0, \quad \forall \theta. \tag{A.5}$$

We give a short proof of Theorem A.1 as a clarification. Readers can also refer to Vincent (2011) or Zhou et al. (2024b) as a reference.

Recall the relation between $\boldsymbol{s}_{p_{\theta,t}}(\boldsymbol{x}_t)$ and $\nabla_{\boldsymbol{x}_t} \log q_t(\boldsymbol{x}_t | \boldsymbol{x}_0)$, we know

$$\boldsymbol{s}_{p_{\theta,t}}(\boldsymbol{x}_t) = \nabla_{\boldsymbol{x}_t} \log \int p_{\theta,0}(\boldsymbol{x}_0) q_t(\boldsymbol{x}_t | \boldsymbol{x}_0) \mathrm{d}\boldsymbol{x}_0$$

$$= \frac{\int p_{\theta,t}(\boldsymbol{x}_0) \nabla_{\boldsymbol{x}_t} q_t(\boldsymbol{x}_t | \boldsymbol{x}_0) \mathrm{d}\boldsymbol{x}_0}{p_{\theta,t}(\boldsymbol{x}_t)}$$

$$= \int \frac{\nabla_{\boldsymbol{x}_t} \log q_t(\boldsymbol{x}_t | \boldsymbol{x}_0) p_{\theta,t}(\boldsymbol{x}_0) q_t(\boldsymbol{x}_t | \boldsymbol{x}_0)}{p_{\theta,t}(\boldsymbol{x}_t)} \mathrm{d}\boldsymbol{x}_0$$

We have

$$\mathbb{E}_{\substack{\boldsymbol{x}_0 \sim p_{\theta,0}, \\ \boldsymbol{x}_t | \boldsymbol{x}_0 \sim q_t(\boldsymbol{x}_t | \boldsymbol{x}_0)}} \boldsymbol{u}(\boldsymbol{x}_t)^T \boldsymbol{s}_{p_{\theta,t}}(\boldsymbol{x}_t) = \mathbb{E}_{\boldsymbol{x}_t \sim p_{\theta,t}} \boldsymbol{u}(\boldsymbol{x}_t)^T \boldsymbol{s}_{p_{\theta,t}}(\boldsymbol{x}_t)$$

$$= \int p_{\theta,t}(\boldsymbol{x}_t) \boldsymbol{u}(\boldsymbol{x}_t)^T \boldsymbol{s}_{p_{\theta,t}}(\boldsymbol{x}_t) \mathrm{d}\boldsymbol{x}_t$$

$$= \int p_{\theta,t}(\boldsymbol{x}_t) \boldsymbol{u}(\boldsymbol{x}_t)^T \int \frac{\nabla_{\boldsymbol{x}_t} \log q_t(\boldsymbol{x}_t | \boldsymbol{x}_0) p_{\theta,t}(\boldsymbol{x}_0) q_t(\boldsymbol{x}_t | \boldsymbol{x}_0)}{p_{\theta,t}(\boldsymbol{x}_t)} \mathrm{d}\boldsymbol{x}_0 \mathrm{d}\boldsymbol{x}_t$$

$$= \int \boldsymbol{u}(\boldsymbol{x}_t)^T \int \nabla_{\boldsymbol{x}_t} \log q_t(\boldsymbol{x}_t | \boldsymbol{x}_0) p_{\theta,t}(\boldsymbol{x}_0) q_t(\boldsymbol{x}_t | \boldsymbol{x}_0) \mathrm{d}\boldsymbol{x}_0 \mathrm{d}\boldsymbol{x}_t$$

$$= \int \int \boldsymbol{u}(\boldsymbol{x}_t)^T \nabla_{\boldsymbol{x}_t} \log q_t(\boldsymbol{x}_t | \boldsymbol{x}_0) p_{\theta,t}(\boldsymbol{x}_0) q_t(\boldsymbol{x}_t | \boldsymbol{x}_0) \mathrm{d}\boldsymbol{x}_0 \mathrm{d}\boldsymbol{x}_t$$

$$= \mathbb{E}_{\substack{\boldsymbol{x}_0 \sim p_{\theta,0}, \\ \boldsymbol{x}_t | \boldsymbol{x}_0 \sim q_t(\boldsymbol{x}_t | \boldsymbol{x}_0)}} \boldsymbol{u}(\boldsymbol{x}_t)^T \nabla_{\boldsymbol{x}_t} \log q_t(\boldsymbol{x}_t | \boldsymbol{x}_0)$$

If we take the $\theta$ gradient on both sides of (A.5), we have

$$0 = \mathbb{E}_{\substack{\boldsymbol{x}_0 \sim p_{\theta,0}, \\ \boldsymbol{x}_t | \boldsymbol{x}_0 \sim q_t(\boldsymbol{x}_t | \boldsymbol{x}_0)}} \left\{ \frac{\partial}{\partial \boldsymbol{x}_t} \left[ \boldsymbol{u}(\boldsymbol{x}_t)^T \left\{ \boldsymbol{s}_{p_{\theta,t}}(\boldsymbol{x}_t) - \nabla_{\boldsymbol{x}_t} \log q_t(\boldsymbol{x}_t | \boldsymbol{x}_0) \right\} \right] \frac{\partial \boldsymbol{x}_t}{\partial \theta} \right.$$

$$\left. - \boldsymbol{u}(\boldsymbol{x}_t)^T \frac{\partial}{\partial \boldsymbol{x}_0} \left[ \nabla_{\boldsymbol{x}_t} \log q_t(\boldsymbol{x}_t | \boldsymbol{x}_0) \right] \frac{\partial \boldsymbol{x}_0}{\partial \theta} \right\} + \mathbb{E}_{\boldsymbol{x}_t \sim p_{\theta,t}} \boldsymbol{u}(\boldsymbol{x}_t)^T \frac{\partial}{\partial \theta} \left\{ \boldsymbol{s}_{p_{\theta,t}}(\boldsymbol{x}_t) \right\} \qquad \text{(A.6)}$$

So we have an identity

$$\mathbb{E}_{\boldsymbol{x}_t \sim p_{\theta,t}} \boldsymbol{u}(\boldsymbol{x}_t)^T \frac{\partial}{\partial \theta} \left\{ \boldsymbol{s}_{p_{\theta,t}}(\boldsymbol{x}_t) \right\} = -\frac{\partial}{\partial \theta} \mathbb{E}_{\substack{\boldsymbol{x}_0 \sim p_{\theta,0}, \\ \boldsymbol{x}_t | \boldsymbol{x}_0 \sim q_t(\boldsymbol{x}_t | \boldsymbol{x}_0)}} \left\{ \boldsymbol{u}(\boldsymbol{x}_t) \left\{ \boldsymbol{s}_{p_{\mathrm{sg}[\theta],t}}(\boldsymbol{x}_t) - \nabla_{\boldsymbol{x}_t} \log q_t(\boldsymbol{x}_t | \boldsymbol{x}_0) \right\} \right\}$$

Notice that the left-hand side of equation (A.7) can be interpreted as the gradient of the loss function when the parameter dependency of the sampling distribution is cut off, i.e.

$$\mathbb{E}_{\boldsymbol{x}_t \sim p_{\theta,t}} \boldsymbol{u}(\boldsymbol{x}_t)^T \frac{\partial}{\partial \theta} \left\{ \boldsymbol{s}_{p_{\theta,t}}(\boldsymbol{x}_t) \right\} = \frac{\partial}{\partial \theta} \mathbb{E}_{\boldsymbol{x}_t \sim p_{\mathrm{sg}[\theta],t}} \left\{ \boldsymbol{u}(\boldsymbol{x}_t)^T \boldsymbol{s}_{p_{\theta,t}}(\boldsymbol{x}_t) \right\} \qquad \text{(A.7)}$$

Therefore we have the final equation

$$\frac{\partial}{\partial \theta} \mathbb{E}_{\boldsymbol{x}_t \sim p_{\mathrm{sg}[\theta],t}} \left\{ \boldsymbol{u}(\boldsymbol{x}_t)^T \boldsymbol{s}_{p_{\theta,t}}(\boldsymbol{x}_t) \right\} = -\frac{\partial}{\partial \theta} \mathbb{E}_{\substack{\boldsymbol{x}_0 \sim p_{\theta,0}, \\ \boldsymbol{x}_t | \boldsymbol{x}_0 \sim q_t(\boldsymbol{x}_t | \boldsymbol{x}_0)}} \left\{ \boldsymbol{u}(\boldsymbol{x}_t) \left\{ \boldsymbol{s}_{p_{\mathrm{sg}[\theta],t}}(\boldsymbol{x}_t) - \nabla_{\boldsymbol{x}_t} \log q_t(\boldsymbol{x}_t | \boldsymbol{x}_0) \right\} \right\}$$

$$\text{(A.8)}$$

which holds for arbitrary function $\boldsymbol{u}(\cdot)$ and parameter $\theta$. If we set

$$\boldsymbol{u}(\boldsymbol{x}_t) = \mathbf{d}'(\boldsymbol{y}_t)$$

$$\boldsymbol{y}_t = \boldsymbol{s}_{p_{\mathrm{sg}[\theta],t}}(\boldsymbol{x}_t) - \boldsymbol{s}_{q_t}(\boldsymbol{x}_t)$$

Then we formally have

$$\frac{\partial}{\partial \theta} \mathbb{E}_{\boldsymbol{x}_t \sim p_{\mathrm{sg}[\theta],t}} \left\{ \mathbf{d}'(\boldsymbol{y}_t) \right\}^T \left\{ \boldsymbol{s}_{p_{\theta,t}}(\boldsymbol{x}_t) \right\}$$

$$= \frac{\partial}{\partial \theta} \mathbb{E}_{\substack{\boldsymbol{x}_0 \sim p_{\theta,0}, \\ \boldsymbol{x}_t | \boldsymbol{x}_0 \sim q_t(\boldsymbol{x}_t | \boldsymbol{x}_0)}} \left\{ -\mathbf{d}'(\boldsymbol{y}_t) \right\}^T \left\{ \boldsymbol{s}_{p_{\theta,t}}(\boldsymbol{x}_t) - \nabla_{\boldsymbol{x}_t} \log q_t(\boldsymbol{x}_t | \boldsymbol{x}_0) \right\} \qquad \text{(A.9)}$$

This means that we can use the $\theta$ gradient of a tractable loss:

$$\mathbb{E}_{\substack{t, \boldsymbol{x}_0 \sim p_{\theta,0}, \\ \boldsymbol{x}_t | \boldsymbol{x}_0 \sim q_t(\boldsymbol{x}_t | \boldsymbol{x}_0)}} w(t) \left\{ -\mathbf{d}'(\boldsymbol{y}_t) \right\}^T \left\{ \boldsymbol{s}_{p_{\theta,t}}(\boldsymbol{x}_t) - \nabla_{\boldsymbol{x}_t} \log q_t(\boldsymbol{x}_t | \boldsymbol{x}_0) \right\} \qquad \text{(A.10)}$$

to replace the wanted $\theta$ gradient (A.3), which can minimize the regularization loss (A.2).

Combining $r(\boldsymbol{x}_0, \boldsymbol{c})$ and (A.10), we have the practical loss

$$\mathcal{L}_{DI*}(\theta) = \mathbb{E}_{\substack{\boldsymbol{z} \sim p_z, \\ \boldsymbol{x}_0 = g_\theta(\boldsymbol{z})}} \left[ -\alpha r(\boldsymbol{x}_0, \boldsymbol{c}) \right. \qquad \text{(A.11)}$$

$$\left. + \mathbb{E}_{\substack{t, \boldsymbol{x}_t | \boldsymbol{x}_0 \\ \sim q_t(\boldsymbol{x}_t | \boldsymbol{x}_0)}} w(t) \left\{ -\mathbf{d}'(\boldsymbol{y}_t) \right\}^T \left\{ \boldsymbol{s}_{p_{\mathrm{sg}[\theta],t}}(\boldsymbol{x}_t) - \nabla_{\boldsymbol{x}_t} \log q_t(\boldsymbol{x}_t | \boldsymbol{x}_0) \right\} \mathrm{d}t \right]$$

$\square$

**Remark A.2.** In practice, most commonly used forward diffusion processes can be expressed as a form of scale and noise addition:

$$\boldsymbol{x}_t = \alpha(t)\boldsymbol{x}_0 + \beta(t)\epsilon, \quad \epsilon \sim \mathcal{N}(\epsilon; \mathbf{0}, \boldsymbol{I}). \tag{A.12}$$

So the term $\boldsymbol{x}_t$ in equation (A.11) can be instantiated as $\boldsymbol{z} \sim p_z$, $\epsilon \sim \mathcal{N}(\epsilon; \mathbf{0}, \boldsymbol{I})$, $\boldsymbol{x}_t = \alpha(t)\boldsymbol{x}_0 + \beta(t)\epsilon$.

### A.2 PROOF OF THEOREM 3.2

*Proof.* Recall the definition of the classifier-free reward (3.9). The negative reward writes

$$-r(\boldsymbol{x}_0, \boldsymbol{c}) = -\mathbb{E}_{t,\boldsymbol{x}_t \sim p_{\theta,t}} w(t) \log \frac{p_{ref}(\boldsymbol{x}_t|t, \boldsymbol{c})}{p_{ref}(\boldsymbol{x}_t|t)}$$

This reward will put a higher reward on those samples that have higher class-conditional probability than unconditional probability, therefore encouraging class-conditional sampling. It is clear that

$$\frac{\partial}{\partial \theta}\left\{ -r(\boldsymbol{x}_0, \boldsymbol{c}) \right\} = -\mathbb{E}_{t,\boldsymbol{x}_t \sim p_{\theta,t}} w(t) \left\{ \nabla_{\boldsymbol{x}_t} \log p_{ref}(\boldsymbol{x}_t|t, \boldsymbol{c}) - \nabla_{\boldsymbol{x}_t} \log p_{ref}(\boldsymbol{x}_t|t) \right\} \frac{\partial \boldsymbol{x}_t}{\partial \theta}$$

$$= -\mathbb{E}_{t,\boldsymbol{x}_t \sim p_{\theta,t}} w(t) \left\{ \boldsymbol{s}_{ref}(\text{sg}[\boldsymbol{x}_t]|t, \boldsymbol{c}) - \boldsymbol{s}_{ref}(\text{sg}[\boldsymbol{x}_t]|t, \varnothing) \right\} \frac{\partial \boldsymbol{x}_t}{\partial \theta} \tag{A.13}$$

Therefore, we can see that the equivalent loss

$$\mathcal{L}_{cfg}(\theta) = \mathbb{E}_{\substack{t,\boldsymbol{z} \sim p_z, \boldsymbol{x}_0 = g_\theta(\boldsymbol{z}|\boldsymbol{c}) \\ \boldsymbol{x}_t|\boldsymbol{x}_0 \sim p(\boldsymbol{x}_t|\boldsymbol{x}_0)}} w(t) \left\{ \boldsymbol{s}_{ref}(\text{sg}[\boldsymbol{x}_t]|t, \boldsymbol{c}) - \boldsymbol{s}_{ref}(\text{sg}[\boldsymbol{x}_t]|t, \varnothing) \right\}^T \boldsymbol{x}_t \tag{A.14}$$

recovers the gradient formula (A.13). $\qquad\square$

## B IMPORTANT MATERIALS FOR MAIN CONTENT

### B.1 PROMPTS FOR FIGURE 1 AND FIGURE 2

Prompts for Figure 1 (from upper left to bottom right):

- *A girl examining an ammonite fossil*;
- *A squirrel driving a toy car*;
- *A portrait of a statue of the Egyptian god Anubis wearing aviator goggles, white t-shirt and leather jacket. The city of Los Angeles is in the background*;
- *A still image of a humanoid cat posing with a hat and jacket in a bar*;
- *A photograph of the inside of a subway train. There are red pandas sitting on the seats. One of them is reading a newspaper. The window shows the jungle in the background*;
- *A capybara made of voxels sitting in a field*;
- *A teddy bear on a skateboard in times square*;
- *A sloth in a go kart on a race track. The sloth is holding a banana in one hand. There is a banana peel on the track in the background*;
- *A close-up photo of a wombat wearing a red backpack and raising both arms in the air*;
- *A small cactus with a happy face in the Sahara desert*;
- *Baker proudly displays her white dog cake in her kitchen*;
- *A bowl with rice, broccoli and a purple relish*;
- *An inlet filled with boats of all kinds*;
- *A black cat sitting on top of the hood of a car*;
- *A woman wearing a cowboy hat face to face with a horse.*

Prompts for Figure 2. The prompts are listed from the up rows to the bottom row:

- *Pirate ship sailing into a bioluminescence sea with a galaxy in the sky, epic, 4k, ultra*;

- *Digital 2D, Miyazaki's style, ultimate detailed, tiny finnest details, futuristic, sci-fi, magical dreamy landscape scenery, small cute girl living alone with plushified friendly big tanuki in the gigantism of wilderness, intricate round futuristic simple multilayered architecture, habitation cabin in the trees, dramatic soft lightning, rule of thirds, cinematic*;

- *saharian landscape at sunset, 4k ultra realism, BY Anton Gorlin, trending on artstation, sharp focus, studio photo, intricate details, highly detailed, by greg rutkowski*.

## B.2    MEANINGS OF HYPER-PARAMETERS.

**Meanings of Hyper-parameters.**    As in Algorithm 1, the overall algorithms consist of two alternative updating steps. The first step is to update $\psi$ of the assistant diffusion model by fine-tuning it with student-generated data. Therefore the assistant diffusion $s_\psi(x_t|t, c)$ can approximate the score function of student generator distribution. This step means that the  assistant diffusion needs to communicate with the student to know the student's status. The second step updates the generator by minimizing the tractable loss (3.7) using SGD-based optimization algorithms such as Adam (Kingma & Ba, 2014). This step means that the teacher and the assistant diffusion discuss and incorporate the student's interests to instruct the student generator.

As we can see in Algorithm 1 (as well as Algorithm 2). Each hyperparameter has its intuitive meaning. The reward scale parameter $\alpha_{rew}$ controls the strength of human preference alignment. The larger the $\alpha_{rew}$ is, the stronger the generator is aligned with human preferences. However, the drawback for a too large $\alpha_{rew}$ might be the loss of diversity and reality. Besides, we empirically find that larger $\alpha_{rew}$ leads to richer generation details and better generation layouts. But a very large $\alpha_{rew}$ results in unrealistic and painting-like images.

The CFG reward scale controls the strength of using CFG rewards when training. We empirically find that the best CFG scale for Diff-Instruct* is the same as the best CFG scale for sampling from the reference diffusion model. However, $\alpha_{cfg}$ may conflicts with $\alpha_{rew}$. In the Stable Diffusion 1.5 experiment, we find that using a large CFG reward scale leads to worse human preferences. Therefore, the proper combination of $(\alpha_{rew}, \alpha_{cfg})$ asks for careful tuning.

The diffusion model weighting $\lambda(t)$ and the generator loss weighting $w(t)$ controls the strengths put on each time level of updating assistant diffusion and the student generator. We empirically find that it is decent to set $\lambda(t)$ to be the same as the default training weighting function for the reference diffusion. And it is decent to set the $w(t) = 1$ for all time-levels in practice. In the following section, we give more discussions on Diff-Instruct*.

## B.3    MORE DISCUSSIONS ON DIFF-INSTRUCT*

**Flexible Choices of Divergences.**    Clearly, various choices of distance function $\mathbf{d}(.)$ result in different training algorithms. In this part, we discuss two instances. The first choice distance function is a simple squared distance, i.e. $\mathbf{d}(y_t) = \|y_t\|_2^2$. The corresponding derivative term writes $\mathbf{d}'(y_t) = 2y_t$. In fact, such a distance function recovers the practical diffusion distillation loss studied in Zhou et al. (2024b;a). The second distance is the pseudo-Huber distance, which shows more robust performances than the simple squared distance. The pseudo-Huber distance is defined with $\boldsymbol{d}(\boldsymbol{y}) \coloneqq \sqrt{\|y_t\|_2^2 + c^2} - c$, where $c$ is a pre-defined positive constant. The corresponding regularization loss (3.6) writes

$$\mathbf{D}^{[0,T]}(p_\theta, p_{ref}) = -\left\{ \frac{y_t}{\sqrt{\|y_t\|_2^2 + c^2}} \right\}^T \left\{ s_\psi(x_t, t) - \nabla_{x_t} \log q_t(x_t|x_0) \right\}. \qquad \text{(B.1)}$$

Here $y_t \coloneqq s_{p_{\mathrm{sg}[\theta],t}}(x_t) - s_{q_t}(x_t)$.

**DI* Does Not Need Image Data When Training.**    One appealing advantage of DI* is the image data-free property, which means that DI* requires neither the image datasets nor synthetic images that are generated by reference diffusion models. This advantage distinguishes DI* from previous fine-tuning methods such as generative adversarial training (Goodfellow et al., 2014) which require training additional neural classifiers over image data, as well as those fine-tuning methods over large-scale synthetic or curated datasets.

Table 3: HPSv2.0 score of SDXL-based 1-step model alignment using Diff-Instruct*. We mark the increment over the initial model with green number to demonstrate its trend. We can see that the increment converges to +0.42.

| K Images | 0 | 102 | 205 | 307 | 410 | 512 | 614 | 717 | 819 | 922 |
|---|---|---|---|---|---|---|---|---|---|---|
| Concept-art↑ | 27.52 | 27.64 | 27.64 | 27.70 | 27.78 | 27.90 | 27.98 | 28.02 | 28.04 | 28.05 |
| Photo↑ | 27.75 | 27.85 | 27.83 | 27.88 | 27.98 | 28.05 | 28.10 | 28.09 | 28.12 | 28.09 |
| Animation↑ | 28.45 | 28.52 | 28.58 | 28.56 | 28.68 | 28.76 | 28.79 | 28.78 | 28.74 | 28.73 |
| Painting↑ | 27.52 | 27.60 | 27.68 | 27.73 | 27.85 | 27.95 | 28.04 | 28.09 | 28.14 | 28.11 |
| Avg HPSv2.0↑ | 27.81 | 27.90(+0.09) | 27.93(+0.11) | 27.97(+0.17) | 28.07(+0.26) | 28.17(+0.36) | 28.23(+0.42) | 28.24(+0.43) | 28.26(+0.05) | 28.24(+0.43) |

**The Choice of Generator is Flexible across Broader Applications.** Another interesting property of DI* for alignment is its wide flexibility in the choice of generator models. We can see that, the theory of DI* only requires the generator to be able to generate output images (or data of other modalities) that are differentiable with the generator's parameters. This makes DI* a universal training method for two reasons. 1) the choice of generator architecture is flexible. The network architectures for diffusion models require the input and output to have the same dimensions. However, the DI* does not assign such a restriction to generator network choices. Therefore, pre-trained GAN generators, such as StyleGAN-T (Sauer et al., 2023a) and GigaGAN (Kang et al., 2023b) are also compatible with DI*. Besides, we have also shown in Section 5.2 that DI* is compatible with both UNet-based and DiT-based generator architectures. 2) student networks in broader applications may also satisfy the requirements of DI*. For instance, the neural radiance field (Mildenhall et al., 2021) model used in text-to-3D generation using text-to-2D diffusion models can also be viewed as a generator. Therefore DI* can be used for such scenarios to incorporate human preference in the training process. Readers can read Poole et al. (2022), Wang et al. (2023) for more introductions.

### B.4 Experiment Details for Pre-training and Alignment

**The human preference score (HPSv2.0) trend of DI*-SDXL-1step model.** During the training of the DI*-SDXL-1step model, we monitor the change of the HPSv2.0 score as an out-of-sample validation metric. We initialize the 1-step model with DMD2-SDXL-1step model (Yin et al., 2024), which is a pretty solid SDXL-based one-step diffusion distillation model. We set the learning rate of both the one-step generator and the online diffusion model to be $1e-5$ and set an exponential moving average decay rate of 0.9 for faster convergence. Following the same setting as SD1.5 and PixelArt-$\alpha$ experiment, we use the ImageReward as the explicit reward, while using a CFG scale of 8.0 for implicit CFG reward. Table 3 records the HPSv2.0 trend of the alignment process using Diff-Instruct*.

**Detailed Experiment Settings for SD1.5 Experiment.** For experiments of SD1.5, we use the open-sourced SD1.5 of a resolution of 512×512 as our reference diffusion in Algorithm 1. We implement our experiments based on SiD-LSG (Zhou et al., 2024a) [1], which provides a high-quality codebase for diffusion model training and distillation. We construct the one-step generator with the same architecture as the reference SD1.5 model, following the same configuration of SiD-LSG. We use the prompts of the LAION-AESTHETIC dataset with an aesthetic score larger than 6.25, which resulting a total of 3M text prompts. We only prepare the text prompts since DI* does not need image datasets. We use the off-the-shelf Image Reward [2] as our explicit reward model. To better explore the advantages of our score-based divergence over traditional KL divergence, we refer to a recent work (Luo, 2024) that uses KL divergence for training and conducts a detailed comparison between score-based divergences that DI* uses and KL divergences in previous works.

**Detailed Experiment Settings for PixelArt-$\alpha$ Experiment.** Different from Stabld Diffusion models, the PixArt-$\alpha$ model is a high-quality open-sourced text-to-image diffusion model. It uses a diffusion transformer (Peebles & Xie, 2022) to learn marginal score functions in a latent space encoded by a down-sampled variational auto-encoder (VAE) (Rombach et al., 2022). For the text conditioning mechanism, the PixelArt-$\alpha$ model uses a T5-XXL text encoder(Raffel et al., 2020; Tay et al., 2021), which makes the model able to understand long prompts without an obvious length

---

[1] https://github.com/mingyuanzhou/SiD-LSG
[2] https://github.com/THUDM/ImageReward

restriction. We use the DiT architecture and the 0.6B PixArt-$\alpha$ of a resolution of $512\times512$ as our reference diffusion to demonstrate the compatibility of DI* for different kinds of neural network architectures. We use the prompts from the SAM-LLaVA-Caption-10M dataset as our prompt dataset. The SAM-LLaVA-Caption-10M dataset contains the images collected by Kirillov et al. (2023), together with text descriptions that are captioned by LLaVA model (Liu et al., 2024). The SAM-LLaVA-Caption-10M dataset is used for training the PixelArt-$\alpha$ model. Since the PixelArt-$\alpha$ diffusion model uses a T5-XXL, which is memory and computationally expensive. To speed up the alignment training, we pre-encoded the text prompts using the T5-XXL text encoder, saved the encoded embedding vectors, and built the data loaders in-house.

We follow the setting of Diff-Instruct (Luo et al., 2024a) to use the same neural network architecture as the reference diffusion model for the one-step generator. The PixelArt-$\alpha$ model is trained using so-called VP diffusion(Song et al., 2020), which first scales the data in the latent space, then adds noise to the scaled latent data. We reformulate the VP diffusion as the form of so-called *data-prediction* proposed in EDM paper (Karras et al., 2022) by re-scaling the noisy data with the inverse of the scale that has been applied to data with VP diffusion. Under the data-prediction formulation, we select a fixed noise $\sigma_{init}$ level to be $\sigma_{init} = 2.5$ following the Diff-Instruct and SiD (Zhou et al., 2024b). For generation, we first generate a Gaussian vector $\boldsymbol{z} \sim p_z = \mathcal{N}(\boldsymbol{0}, \sigma_{init}^2\mathbf{I})$. Then we input $\boldsymbol{z}$ into the generator to generate the latent. The latent vector can then be decoded by the VAE decoder to turn into an image if needed.

We put the details of how to construct the one-step generator in the following paragraphs. We also initialize the assistant diffusion model with the same weight as the reference diffusion. We use the Image Reward as the human preference reward and use the Diff-Instruct* algorithm 1 (or equivalently the algorithm 2) to train the generator. We also used the Adam optimizer with the parameter $(\beta_1, \beta_2) = (0.0, 0.999)$ for both the generator and the assistant diffusion with a batch size of 128, implemented with BF16 numerical format and the accumulate-gradient training technique. We use a fixed exponential moving average decay (EMA) rate of 0.95 for all training trials. After the training, the generator aligned with both strong CFG and reward model shows significantly improved aesthetic appearance, better generation layout, and richer image details. Figure 2 shows a demonstration of the generated images using our aligned one-step generator with a CFG scale $\alpha_{cfg}$ of 4.5 and a reward scale $\alpha_{rew}$ of 10.0.

**Construction of the one-step generator.** We follow the experiment setting of Diff-Instruct (Luo et al., 2024b), generalizing its CIFAR10 experiment to text-to-image generation. Notice that the Diff-Instruct uses the EDM model (Karras et al., 2022) to formulate the diffusion model, as well as the one-step generator. We start with a brief introduction to the EDM model.

The EDM model depends on the diffusion process

$$\mathrm{d}\boldsymbol{x}_t = t\mathrm{d}\boldsymbol{w}_t, t \in [0, T]. \tag{B.2}$$

Samples from the forward process (B.2) can be generated by adding random noise to the output of the generator function, i.e., $\boldsymbol{x}_t = \boldsymbol{x}_0 + t\boldsymbol{\epsilon}$ where $\boldsymbol{\epsilon} \sim \mathcal{N}(\boldsymbol{0}, \boldsymbol{I})$ is a Gaussian vector. The EDM model also reformulates the diffusion model's score matching objective as a denoising regression objective, which writes,

$$\mathcal{L}(\psi) = \int_{t=0}^{T} \lambda(t)\mathbb{E}_{\boldsymbol{x}_0 \sim p_0, \boldsymbol{x}_t | \boldsymbol{x}_0 \sim p_t(\boldsymbol{x}_t | \boldsymbol{x}_0)}\|\boldsymbol{d}_\psi(\boldsymbol{x}_t, t) - \boldsymbol{x}_0\|_2^2\mathrm{d}t. \tag{B.3}$$

Where $\boldsymbol{d}_\psi(\cdot)$ is a denoiser network that tries to predict the clean sample by taking noisy samples as inputs. Minimizing the loss (B.3) leads to a trained denoiser, which has a simple relation to the marginal score functions as:

$$\boldsymbol{s}_\psi(\boldsymbol{x}_t, t) = \frac{\boldsymbol{d}_\psi(\boldsymbol{x}_t, t) - \boldsymbol{x}_t}{t^2} \tag{B.4}$$

Under such a formulation, we actually have pre-trained denoiser models for experiments. Therefore, we use the EDM notations in later parts.

Let $\boldsymbol{d}_\theta(\cdot)$ be pretrained EDM denoiser models. Owing to the denoiser formulation of the EDM model, we construct the generator to have the same architecture as the pre-trained EDM denoiser with a pre-selected index $t^*$, which writes

$$\boldsymbol{x}_0 = g_\theta(\boldsymbol{z}) := \boldsymbol{d}(\boldsymbol{z}, t^*), \;\; \boldsymbol{z} \sim \mathcal{N}(\boldsymbol{0}, (t^*)^2\mathbf{I}). \tag{B.5}$$

We initialize the generator with the same parameter as the teacher EDM denoiser model.

**Time index distribution.** When training both the EDM diffusion model and the generator, we need to randomly select a time $t$ in order to approximate the integral of the loss function (B.3). The EDM model has a default choice of $t$ distribution as log-normal when training the diffusion (denoiser) model, i.e.

$$t \sim p_{EDM}(t): \quad t = \exp(s) \tag{B.6}$$

$$s \sim \mathcal{N}(P_{mean}, P_{std}^2), \quad P_{mean} = -2.0, P_{std} = 2.0. \tag{B.7}$$

And a weighting function

$$\lambda_{EDM}(t) = \frac{(t^2 + \sigma_{data}^2)}{(t \times \sigma_{data})^2}. \tag{B.8}$$

In our algorithm, we follow the same setting as the EDM model when updating the online diffusion (denoiser) model.

**Weighting function.** For the assistant diffusion updates in both pre-training and alignment, we use the same $\lambda_{EDM}(t)$ (B.8) weighting function as EDM when updating the denoiser model. When updating the generator, we use a specially designed weighting function, which writes:

$$w_{Gen}(t) = \frac{1}{\|\boldsymbol{d}_\psi(\text{sg}[\boldsymbol{x}_t], t) - \boldsymbol{d}_{q_t}(\text{sg}[\boldsymbol{x}_t], t)\|_2} \tag{B.9}$$

$$\boldsymbol{x}_t = \boldsymbol{x}_0 + t\epsilon, \quad \epsilon \sim \mathcal{N}(\boldsymbol{0}, \mathbf{I}) \tag{B.10}$$

The notation $\text{sg}[\cdot]$ means stop-gradient of the parameter. Such a weighting function helps to stabilize the training.

In the both Stable Diffusion 1.5 and the PixArt-$\alpha$ experiments, we rewrite the PixelArt-$\alpha$ model in EDM formulation:

$$D_\theta(\mathbf{x}; \sigma) = \mathbf{x} - \sigma F_\theta \tag{B.11}$$

Here, following the iDDPM+DDIM preconditioning in EDM, PixelArt-$\alpha$ is denoted by $F_\theta$, $\mathbf{x}$ is the image data plus noise with a standard deviation of $\sigma$, for the remaining parameters such as $C_1$ and $C_2$, we kept them unchanged to match those defined in EDM. Unlike the original model, we only retained the image channels for the output of this model. Since we employed the preconditioning of iDDPM+DDIM in the EDM, each $\sigma$ value is rounded to the nearest 1000 bins after being passed into the model. For the actual values used in PixelArt-$\alpha$, beta_start is set to 0.0001, and beta_end is set to 0.02. Therefore, according to the formulation of EDM, the range of our noise distribution is [0.01, 156.6155], which will be used to truncate our sampled $\sigma$. For our one-step generator, it is formulated as:

$$g_\theta(\mathbf{x}; \sigma_{\text{init}}) = \mathbf{x} - \sigma_{\text{init}} F_\theta \tag{B.12}$$

Here following Diff-Instruct to use $\sigma_{\text{init}} = 2.5$ and $\mathbf{x} \sim \mathcal{N}(0, \sigma_{\text{init}}\mathbf{I})$, we observed in practice that larger values of $\sigma_{\text{init}}$ lead to faster convergence of the model, but the difference in convergence speed is negligible for the complete model training process and has minimal impact on the final results.

**Detailed Quantitative Evaluations Metrics** To quantitatively evaluate the performances of the generators trained with different alignment settings, we compare our generators elaborately with other open-sourced models that are either based on SD1.5 diffusion models or larger models such as SDXL. For all models, we compute four standard scores: the Image Reward (Xu et al., 2023a), the Aesthetic Score (Schuhmann, 2022), the PickScore(Kirstain et al., 2023), and the CLIP score(Radford et al., 2021). Since most existing literature tests the human preference scores with different prompts which are possibly not available, to make a fair comparison, in our experiment, we fix 1k prompts from the COCO-2017 (Lin et al., 2014) validation dataset and intensively evaluate a wide range of existing open-sourced models as SiD-LSG (Zhou et al., 2024a), SDXL-Turbo (Sauer et al., 2023b), Latent Consistency Model (LCM) (Luo et al., 2023a), Hyper-SD (Ren et al., 2024), SDXL-Lightning (Lin et al., 2024), Trajectory Consistency Model (TCD) (Zheng et al., 2024), PeReflow (Yan et al., 2024), InstaFLow(Liu et al., 2023), Diffusion DPO(Wallace et al., 2024), etc. All models are tested with the same prompts and the same computing devices.

Besides the COCO prompts, we also evaluate generators with open-sourced models with Human Preference Score v2.0 (HPSv2.0) (Wu et al., 2023) over their benchmark prompts. The HPS is a

widely used standard benchmark that evaluates models' capability of generating images of 4 styles: Animation, concept art, Painting, and Photo. The score reflects the prompt following and the human preference strength of text-to-image models. We use the HPSv2's [3] default protocols for evaluations. Since our generators based on PixArt-$\alpha$ are trained with SAM-recaptioned datasets, we also evaluate the performances on 30K prompts from the SAM-recap dataset.

**Sample Prompts from COCO validation dataests.** In this paragraph, we propose some sample prompts from the COCO validation dataset that we use for calculating scores like Image Reward and the Aesthetic Scores.

```
1  This wire metal rack holds several pairs of shoes and sandals
2  A motorcycle parked in a parking space next to another motorcycle.
3  A picture of a dog laying on the ground.
4  A loft bed with a dresser underneath it.
5  Two giraffes in a room with people looking at them.
6  A woman stands in the dining area at the table.
7  Birds perch on a bunch of twigs in the winter.
8  A small kitchen with low a ceiling
9  A group of baseball players is crowded at the mound.
10 This table is filled with a variety of different dishes.
11 A toy dinosaur standing on a sink next to a running faucet.
12 a man standing holding a game controller and two people sitting
13 There is a small bus with several people standing next to it.
14 A bottle on wine next to a glass of wine.
15 A big burly grizzly bear is show with grass in the background.
16 A man standing in front of a microwave next to pots and pans.
17 Three men in military suits are sitting on a bench,
18 Two people standing in a kitchen looking around.
19 A group of men playing a game of baseball on top of a baseball field.
20 A traffic light over a street surrounded by tall buildings.
21 The snowboarder has jumped high into the air from a snow ramp.
22 A smart phone with an image of a person on it's screen.
23 A man talking on his phone in the public.
24 A cheesy pizza sitting on top of a table.
25 A dog sitting on the inside of a white boat.
26 A guy jumping with a tennis racket in his hand.
27 a close up of a child next to a cake with balloons
28 A man holding a camera up over his left shoulder.
29 A plane flies over water with two islands nearby.
30 A young boy getting ready to catch a baseball in a grass field.
```

Listing 1: Example Prompts from COCO validation dataset.

### B.5 MORE DISCUSSIONS ON FINDINGS OF QUALITATIVE EVALUATIONS

There are some other interesting findings when qualitatively evaluate different models.

- First, we find that the images generated by the aligned model show a better composition when organizing the contents presented in the image. For instance, the main objects of the generated image are smaller and show a more natural layout than other models, with the objects and the background iterating aesthetically. This in turn reveals human preference: human beings would prefer that the object of an image does not take up all spaces of an image;

- Second, we find that the aligned model has richer details than the unaligned model. The stronger we align the model, the richer details the model will generate. Sometimes these rich details come as a hint to the readers about the input prompts. Sometimes they just come to improve the aesthetic performance. We think this phenomenon may be caused by the fact that human prefers images with rich details. Another finding is that as the reward scale for alignment becomes stronger, the generated image from the alignment model becomes more colorful and more similar to paintings. Sometimes this leads to a loss of reality to some

---

[3] https://github.com/tgxs002/HPSv2

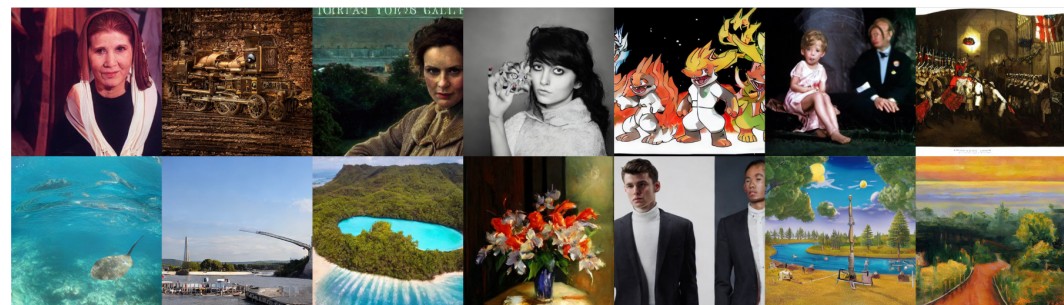

**Before human-preference alignment**

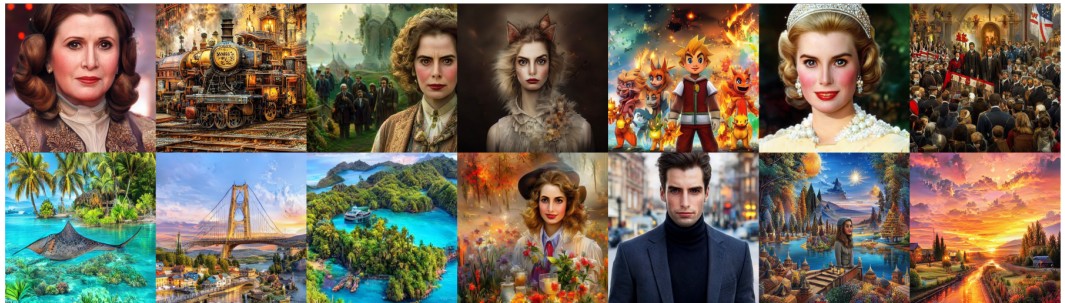

**After human-preference alignment with DI***

Figure 4: A visual comparison of the 0.68B SD1.5-DI* model before (initialized with SiD-LSG pre-trained weights) and after alignments with DI*. We put the prompt in Appendix B.1.

---

**Algorithm 2:** Diff-Instruct* Pseudo Code.

---

**Input:** prompt dataset $\mathcal{C}$, generator $g_\theta(\boldsymbol{x}_0|\boldsymbol{z}, \boldsymbol{c})$, prior distribution $p_z$, reward model $r(\boldsymbol{x}, \boldsymbol{c})$, reward model scale $\alpha_{rew}$, CFG reward scale $\alpha_{cfg}$, reference diffusion model $\boldsymbol{s}_{ref}(\boldsymbol{x}_t|c, \boldsymbol{c})$, assistant diffusion $\boldsymbol{s}_\psi(\boldsymbol{x}_t|t, \boldsymbol{c})$, forward diffusion $p(\boldsymbol{x}_t|\boldsymbol{x}_0)$ (2.1), assistant diffusion updates rounds $K_{TA}$, time distribution $\pi(t)$, diffusion model weighting $\lambda(t)$, generator IKL loss weighting $w(t)$.

**while** *not converge* **do**

    freeze $\theta$, update $\psi$ for $K_{TA}$ rounds by

        1. sample prompt $\boldsymbol{c} \sim \mathcal{C}$; sample time $t \sim \pi(t)$; sample $\boldsymbol{z} \sim p_z(\boldsymbol{z})$;

        2. generate fake data: $\boldsymbol{x}_0 = \text{sg}[g_\theta(\boldsymbol{z}, \boldsymbol{c})]$; sample noisy data: $\boldsymbol{x}_t \sim p_t(\boldsymbol{x}_t|\boldsymbol{x}_0)$;

        3. update $\psi$ by minimizing loss: $\mathcal{L}(\psi) = \lambda(t)\|\boldsymbol{s}_\psi(\boldsymbol{x}_t|t, \boldsymbol{c}) - \nabla_{\boldsymbol{x}_t} \log p_t(\boldsymbol{x}_t|\boldsymbol{x}_0)\|_2^2$;

    freeze $\psi$, update $\theta$ using SGD:

        1. sample prompt $\boldsymbol{c} \sim \mathcal{C}$; sample time $t \sim \pi(t)$; sample $\boldsymbol{z} \sim p_z(\boldsymbol{z})$;

        2. generate fake data: $\boldsymbol{x}_0 = g_\theta(\boldsymbol{z}, \boldsymbol{c})$; sample noisy data: $\boldsymbol{x}_t \sim p_t(\boldsymbol{x}_t|\boldsymbol{x}_0)$;

        3. explicit reward: $\mathcal{L}_{rew}(\theta) = -\alpha_{rew} r(\boldsymbol{x}_0, \boldsymbol{c})$;

        4. CFG reward: $\mathcal{L}_{cfg}(\theta) = \alpha_{cfg} \cdot w(t)\big\{\boldsymbol{s}_{ref}(\text{sg}[\boldsymbol{x}_t]|t, \boldsymbol{c}) - \boldsymbol{s}_{ref}(\text{sg}[\boldsymbol{x}_t]|t, \varnothing)\big\}^T \boldsymbol{x}_t$;

        5. score-regularization:

        $\mathcal{L}_{reg}(\theta) = -w(t)\big\{\mathbf{d}'(\boldsymbol{s}_\psi(\boldsymbol{x}_t|t, \boldsymbol{c}) - \boldsymbol{s}_{ref}(\boldsymbol{x}_t|t, \boldsymbol{c}))\big\}^T \big\{\boldsymbol{s}_\psi(\boldsymbol{x}_t|t, \boldsymbol{c}) - \nabla_{\boldsymbol{x}_t} \log p_t(\boldsymbol{x}_t|\boldsymbol{x}_0)\big\}$;

        6. update $\theta$ by minimizing DI* loss: $\mathcal{L}_{DI*}(\theta) = \mathcal{L}_{rew}(\theta) + \mathcal{L}_{cfg}(\theta) + \mathcal{L}_{reg}(\theta)$;

**end**

**return** $\theta, \psi$.

---

degree. Therefore, we think that users should choose different aligned one-step models with a trade-off between aesthetic performance and image reality according to the use case.

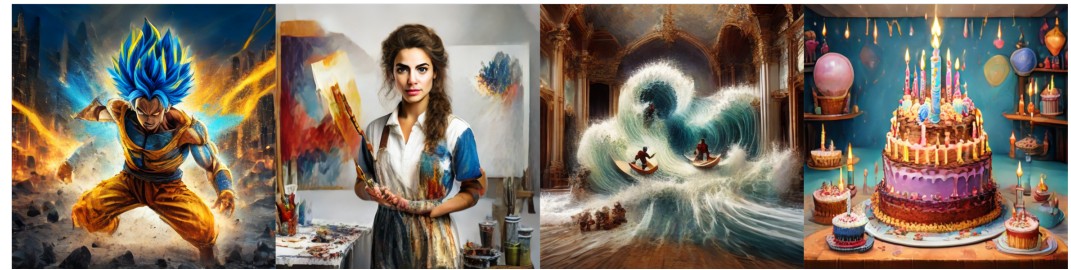

Figure 5: Bad generation cases by aligned DiT-DI* one-step generator model (4.5 CFG + 10.0 reward).

Table 4: Hyperparameters used for Diff-Instruct* on SD1.5 and PixArt-$\alpha$ experiments.

| Hyperparameter | SD1.5 Experiment | | PixArt-$\alpha$ Experiment | |
|---|---|---|---|---|
| | DM $s_\psi$ | Generator $g_\theta$ | DM $s_\psi$ | Generator $g_\theta$ |
| Learning rate | 1e-5 | 1e-5 | 2e-6 | 2e-6 |
| Batch size | 512 | 512 | 256 | 256 |
| $\sigma(t^*)$ | 2.5 | 2.5 | 2.5 | 2.5 |
| $Adam\ \beta_0$ | 0.0 | 0.0 | 0.0 | 0.0 |
| $Adam\ \beta_1$ | 0.999 | 0.999 | 0.999 | 0.999 |
| EMA decay rate | 0.9 | 0.9 | 0.95 | 0.95 |
| Time Distribution | $p_{EDM}(t)$(B.6) | $p_{EDM}(t)$(B.6) | $p_{EDM}(t)$(B.6) | $p_{EDM}(t)$(B.6) |
| Weighting | $\lambda_{EDM}(t)$(B.8) | 1 | $\lambda_{EDM}(t)$(B.8) | 1 |
| Number of GPUs | 8×H800-80G | 8×H800-80G | 8×H100-80G | 8×H800-80G |

With the optimal setting and EDM formulation, we can rewrite our algorithm in an EDM style in Algorithm 3.

---

**Algorithm 3:** Diff-Instruct* Pseudo Code under EDM formulation.

---

**Input:** prompt dataset $\mathcal{C}$, generator $g_\theta(\boldsymbol{x}_0|\boldsymbol{z}, \boldsymbol{c})$, prior distribution $p_z$, reward model $r(\boldsymbol{x}, \boldsymbol{c})$, reward scale $\alpha_{rew}$, CFG scale $\alpha_{cfg}$, reference EDM denoiser model $\boldsymbol{d}_{ref}(\boldsymbol{x}_t|c, \boldsymbol{c})$, assistant EDM denoiser $\boldsymbol{d}_\psi(\boldsymbol{x}_t|t, \boldsymbol{c})$, forward diffusion $p(\boldsymbol{x}_t|\boldsymbol{x}_0)$ (2.1), assistant EDM denoiser updates rounds $K_{TA}$, time distribution $\pi(t)$, diffusion model weighting $\lambda(t)$, generator IKL loss weighting $w(t)$.

**while** *not converge* **do**

    fix $\theta$, update $\psi$ for $K_{TA}$ rounds by

        1. sample prompt $\boldsymbol{c} \sim \mathcal{C}$; sample time $t \sim \pi(t)$; sample $\boldsymbol{z} \sim p_z(\boldsymbol{z})$;

        2. generate fake data: $\boldsymbol{x}_0 = \text{sg}[g_\theta(\boldsymbol{z}, \boldsymbol{c})]$; sample noisy data: $\boldsymbol{x}_t \sim p_t(\boldsymbol{x}_t|\boldsymbol{x}_0)$;

        3. update $\psi$ by minimizing loss: $\mathcal{L}(\psi) = \lambda(t)\|\boldsymbol{d}_\psi(\boldsymbol{x}_t|t, \boldsymbol{c}) - \boldsymbol{x}_0\|_2^2$;

    fix $\psi$, update $\theta$ using StaD:

        1. sample prompt $\boldsymbol{c} \sim \mathcal{C}$; sample time $t \sim \pi(t)$; sample $\boldsymbol{z} \sim p_z(\boldsymbol{z})$;

        2. generate fake data: $\boldsymbol{x}_0 = g_\theta(\boldsymbol{z}, \boldsymbol{c})$; sample noisy data: $\boldsymbol{x}_t \sim p_t(\boldsymbol{x}_t|\boldsymbol{x}_0)$;

        3. explicit reward: $\mathcal{L}_{rew}(\theta) = -\alpha_{rew}r(\boldsymbol{x}_0, \boldsymbol{c})$;

        4. CFG reward: $\mathcal{L}_{cfg}(\theta) = \alpha_{cfg} \cdot w(t)\big\{\boldsymbol{d}_{ref}(\text{sg}[\boldsymbol{x}_t]|t, \boldsymbol{c}) - \boldsymbol{d}_{ref}(\text{sg}[\boldsymbol{x}_t]|t, \varnothing)\big\}^T \boldsymbol{x}_t$;

        5. score-regularization:

$$\mathcal{L}_{reg}(\theta) = -w(t)\big\{\mathbf{d}'(\boldsymbol{d}_\psi(\boldsymbol{x}_t|t, \boldsymbol{c}) - \boldsymbol{d}_{ref}(\boldsymbol{x}_t|t, \boldsymbol{c}))\big\}^T \big\{\boldsymbol{d}_\psi(\boldsymbol{x}_t|t, \boldsymbol{c}) - \boldsymbol{x}_0\big\};$$

        6. update $\theta$ by minimizing DI* loss: $\mathcal{L}_{DI*}(\theta) = \mathcal{L}_{rew}(\theta) + \mathcal{L}_{cfg}(\theta) + \mathcal{L}_{reg}(\theta)$;

**end**

**return** $\theta, \psi$.

---

### B.6 PYTORCH STYLE PSEUDO-CODE OF SCORE IMPLICIT MATCHING

In this section, we give a PyTorch style pseudo-code for algorithm 3.

```python
import torch
import torch.nn as nn
import torch.optim as optim
import copy

use_cfg = True
use_reward = True

# Initialize generator G
G = Generator()

## load teacher DM
Drf = DiffusionModel().load('/path_to_ckpt').eval().requires_grad_(False)
Dta = copy.deepcopy(Drf) ## initialize online DM with teacher DM
r = RewardModel() if use_reward else None

# Define optimizers
opt_G = optim.Adam(G.parameters(), lr=0.001, betas=(0.0, 0.999))
opt_Sta = optim.Adam(Dta.parameters(), lr=0.001, betas=(0.0, 0.999))

# Training loop
while True:
    ## update Dta
    Dta.train().requires_grad_(True)
    G.eval().requires_grad_(False)

    ## update assistant diffusion
    prompt = batch['prompt']
    z = torch.randn((1024, 4, 64, 64), device=G.device)
    with torch.no_grad():
```

```
31          fake_x0 = G(z,prompt)
32
33      sigma = torch.exp(2.0*torch.randn([1,1,1,1], device=fake_x0.device) -
        2.0)
34      noise = torch.randn_like(fake_x0)
35      fake_xt = fake_x0 + sigma*noise
36      pred_x0 = Dta(fake_xt, sigma, prompt)
37
38      weight = compute_diffusion_weight(sigma)
39
40      batch_loss = weight * (pred_x0 - fake_x0)**2
41      batch_loss = batch_loss.sum([1,2,3]).mean()
42
43      optimizer_Dta.zero_grad()
44      batch_loss.backward()
45      optimizer_Dta.step()
46
47
48      ## update G
49      Dta.eval().requires_grad_(False)
50      G.train().requires_grad_(True)
51
52      prompt = batch['prompt']
53      z = torch.randn((1024, 4, 64, 64), device=G.device)
54      fake_x0 = G(z, prompt)
55
56      sigma = torch.exp(2.0*torch.randn([1,1,1,1], device=fake_x0.device) -
        2.0)
57      noise = torch.randn_like(fake_x0)
58      fake_xt = fake_x0 + sigma*noise
59
60      with torch.no_grad():
61          if use_cfg:
62              cfg_vector = (Drf(fake_xt, sigma, prompt) - Drf(fake_xt,
        sigma, None)
63          else:
64              cfg_vector = None
65
66          pred_x0_rf = Drf(fake_xt, sigma, prompt)
67          pred_x0_ta = Dta(fake_xt, sigma, prompt)
68
69      denoise_diff = pred_x0_ta - pred_x0_rf
70      adp_wgt = torch.sqrt(denoise_diff.square().sum([1,2,3], keepdims=True
        ) + phuber_c**2)
71      weight = compute_G_weight(sigma, denoise_diff)
72
73      # compute score regularization loss
74      batch_loss = weight * denoise_diff * (fake_D_yn - D_yn)/adp_wgt
75
76      # compute explicit reward loss if needed
77      if use_reward:
78          reward_loss = -reward_scale * r(fake_x0, prompt)
79          batch_loss += reward_loss
80
81      # compute cfg reward loss if needed
82      if use_cfg:
83          cfg_reward_loss = cfg_scale * cfg_vector*fake_x0
84          batch_loss += cfg_reward_loss
85
86      batch_loss = batch_loss.sum([1,2,3]).mean()
87
88      optimizer_G.zero_grad()
89      batch_loss.backward()
```

```
90      optimizer_G.step()
```

Listing 2: Pytorch Style Pseudo-code of Diff-Instruct*

