# Supplementary Materials for Diff-Instruct*

The used prompts for Figure 1 (from the top row to the bottom row):

- *The image features close-up portrait of a face with big eyes, overflowing with a mysterious, ethereal ambiance, capturing a direct gaze with the viewer amidst the chiaroscuro interplay of light and shadow, featuring soft colors alongside a splash of color from variously hued flowers, all set against a green retro Gothic night scene reminiscent of Mr X's style, face illuminated, reflective surfaces enhancing the rich, vivid textures, and the delicate details enhanced by octane rendering, cinematic portrayal.*

- *The image features seasoned fisherman portrait, weathered skin etched with deep wrinkles, white beard, piercing gaze beneath a fisherman's hat, softly blurred dock background accentuating rugged features, captured under natural light, ultra-realistic, high dynamic range photo.*

- *A serene meadow with a tree, river, bridge, and mountains in the background under a slightly overcast sunrise sky.*

054
055
056
057
058
059
060
061
062
063
064
065
066
067
068
069
070
071
072
073
074
075
076
077
078
079
080
081
082
083
084
085
086
087
088
089
090
091
092
093
094
095
096
097
098
099
100
101
102
103
104
105
106
107

**Before Alignment (DMD2-SDXL-1step)**  **After Alignment (DI∗-SDXL-1step)**

Figure 1: A demonstration of alignment effects of Diff-Instruct* (Please zoom in to check the details). **Left Column**: the initial one-step model based on SDXL architecture (i.e., DMD2-SDXL-1step); **Right Column**: one-step model aligned with Diff-Instruct* based on SDXL architecture (i.e. DI*-SDXL-1step). **The Diff-Instruct* alignment significantly improves the image quality, resulting in a better layout, richer details, and aesthetic colors.**

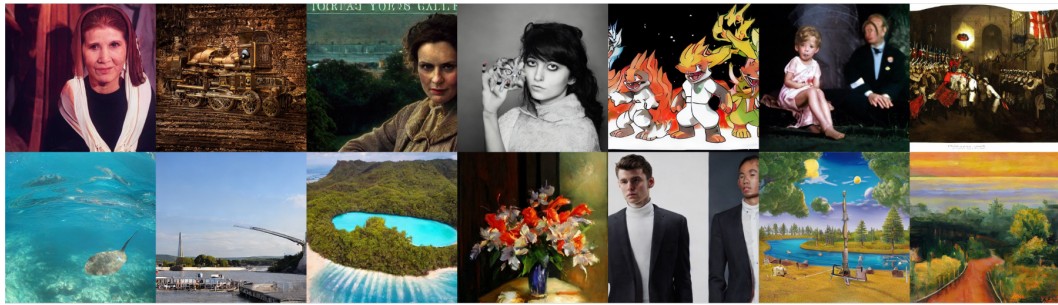

**Before human-preference alignment**

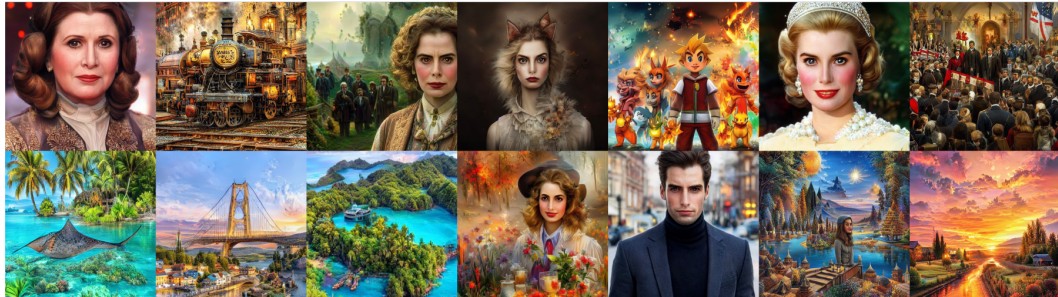

**After human-preference alignment with DI\***

Figure 2: A visual comparison of the 0.68B SD1.5-DI* model before (initialized with SiD-LSG pre-trained weights) and after alignments with DI*. Clearly, the alignment stage using Diff-Instruct* significantly improves the visual quality and aesthetic preferences of the generated images.