# OpenReview forum: "Diff-Instruct*: Towards Human-Preferred One-step Text-to-image Generative Models"
_ICLR.cc/2025/Conference — Submitted to ICLR 2025_

### Official Review · Reviewer_3Wi1 · 2024-10-27

**Soundness:** 4
**Presentation:** 2
**Contribution:** 3
**Rating:** 6
**Confidence:** 5

**Summary:**

This work proposes a method to train a one-step text-to-image generation model that aligns with human preferences. It utilizes reinforcement learning to maximize expected human reward functions with score-based divergence regularization. The proposed approach achieves new state-of-the-art performance on human preference.

**Strengths:**

- This work presents solid and detailed experiments along with thorough theoretical evidence.
- The proposed method achieves a new state-of-the-art performance in both Aesthetic Score and Image Reward.

**Weaknesses:**

- About Quantitative Comparisons: In Table 1, despite the promising Image Reward and Aesthetic Score that have been achieved, the PICK score and CLIP score drop slightly compared to the baseline methods (both SD15-BASE and PIXELART-$\alpha$-512 at different sampling steps). Some further concerns about the CLIP metric are discussed in the Questions and Discussions section.
- The presentation of this paper is incomplete. For example, the caption for Figure 4 mentions that prompts will be provided in Appendix B1, but Appendix B1 does not contain such prompts.

**Questions:**

I thank the authors for their efforts in this work. I have some questions for discussion about this work.

- The elucidation of the evaluation metric (CLIP score) is unclear. As far as I know, it includes two aspects: image-image similarity and image-text similarity. The CLIP score used in Table 1 requires further clarification.

---

### Official Review · Reviewer_98wH · 2024-11-04

**Soundness:** 3
**Presentation:** 4
**Contribution:** 2
**Rating:** 6
**Confidence:** 4

**Summary:**

This paper improves the one-step generation results by transforming the constraint function of RLHF into a diffusion-based form and also introduces CFG reward.

**Strengths:**

1. This paper demonstrates that the constraint function used in RLHF can be implemented using a reference diffusion model, thereby ensuring better results.
2. The proposed DI* introduces the flexible choice of reference diffusion models and generator architecture.

**Weaknesses:**

1. The method of using diffusion as a reference model to ensure the plausibility of one-step generation is not novel. From another perspective, the approach proposed in this paper is more akin to adding a reward loss to existing one-step distillation methods, thus casting doubt on its novelty.
2. The experiments conducted in this paper, which compare using metrics such as IMAGE REWARD against non-RLHF models, are unfair. The baseline for comparison in this paper should be existing one-step generation methods, such as DMD2, with the results after incorporating reward loss.
3. This paper only conducted experiments on proprietary models DiT and SD1.5, which is different from other existing work that was done on SDXL. This lack of parity in comparison makes the conclusions drawn from the model less solid.
4. If the authors provide more fair comparative results, additional ablation study outcomes, and conduct more analyses to demonstrate the effectiveness of their method on the generated results, I would consider raising my rating.

**Questions:**

Please refer to the "Weaknesses" section. If the

---

### Official Review · Reviewer_d57R · 2024-11-04

**Soundness:** 3
**Presentation:** 3
**Contribution:** 2
**Rating:** 6
**Confidence:** 4

**Summary:**

This paper proposes an approach to simultaneously perform diffusion distillation and human preference alignment to create a one-step human-preferred text-to-image generation model. The authors combine reward optimization loss with diffusion distillation loss, incorporating an auxiliary classifier-free guidance-based implicit reward optimization. These components work in tandem to achieve good final performance. The method is evaluated on Unet-based and transformer-based diffusion models, demonstrating solid empirical results on MSCOCO validation prompts.

**Strengths:**

1. The motivation for developing a one-step image generation model that aligns with human preferences is clear and compelling.

2. The paper introduces an interesting combination of explicit and implicit reward optimization alongside diffusion distillation.

3. The empirical results are promising, with large-scale experiments conducted on Stable Diffusion and PixelArt-Alpha models.

**Weaknesses:**

1. **Data-Free Claim**:
   The claim of being "data-free" is somewhat overstated, as training still requires text prompts. Using different sets of text prompts could potentially lead to varying performance, which undermines the data-free assertion.

2. **Complex Notation**:
   The notation used in the paper is complex and could benefit from simplification for clarity. The reference model and $q(x)$ are sometimes ambiguously interchanged such as the $s_{q_t}$ in (3.4). And $p_t$ appears abruptly in Equation 3.7. This can disrupt reader comprehension.

3. **Optimization of Equation 3.5**:
   It is unclear why Equation 3.5 cannot be directly optimized. The paper introduces the assistant diffusion model $ \psi $, which approximates the score estimate of $ p_{\theta} $. This should allow for direct optimization of Equation 3.4.

4. **Algorithm Name Explain**:
   By looking into the reference papers Diff-Instruct and SiD-LSG, the score-based divergence loss used for diffusion distillation closely resembles the SiD loss from prior work, while not the one used in Diff-Instruct. This raises the question of why the algorithm is named ‘Diff-Instruct*’.

5. **Ablation Study**:
   While the paper proposes a combination of three losses—explicit reward optimization, score-based divergence regularization, and implicit reward optimization—there is a need for a detailed ablation study to show the contribution of each component. It would be particularly insightful to understand the impact of the implicit reward optimization on the final performance.

6. **Evaluation Metrics and Test Sets**:
   The choice of MSCOCO-2017 validation set for testing raises concerns, as it is not a common benchmark for evaluating image preference learning. More standard test sets such as HPSv2, Parti prompts, or Pick-a-Pic are typically used. Additionally, baseline methods like Diffusion-DPO leverage text-image pairs for training, and the choice of test set can significantly impact performance. The simple and short MSCOCO prompts might underrepresent the performance of advanced methods like LCM, DPO, DMD, and SDXL. A comparison using standard test sets would strengthen the evaluation.

**Questions:**

See Weaknesses.

---

### Official Review · Reviewer_Fo2e · 2024-11-06

**Soundness:** 2
**Presentation:** 3
**Contribution:** 3
**Rating:** 5
**Confidence:** 3

**Summary:**

This paper proposes a method to align one-step diffusion models with human preferences. When optimizing for human preferences, a novel score-based divergence regularization is applied so that the fine-tuned model does not drift too much from the original diffusion model. The authors argue that the score-based divergence regularization is better than the KL divergence regularization in prior works. The authors propose a tractable algorithm to compute the gradient of the score-based divergence regularization, based on the theorems they derive.

**Strengths:**

- The proposed DI* achieves higher human preference scores than the baselines by a large margin.
- The proposed score-based divergence regularization is novel and could be potentially applied to other scenarios.
- The theorems look plausible, although I did not verify them.

**Weaknesses:**

- The motivation is not convincing enough. In line 191, the authors suggest the mode-seeking behavior of the KL divergence (KLD) can potentially lead to unstable training dynamics. However, I cannot see why the KLD can be a problem. Can the authors provide more details on this motivation?
- In Figure 3, the curves do not seem to converge. For a rigorous comparison, the models should be trained until convergence, as the baseline may perform better upon convergence.
- In Table 1, the authors should include SiD-LSG as a baseline, as their model is initialized from SiD-LSG. It can also see any improvement of DI* over SiD-LSG.
- Some standard scores like FID and IS should be reported. It can help us understand the aspects of the image quality other than the human preference metrics. The ImageReward and PickScore focus much more on text-image alignment (how well the image is aligned with the prompt) than the image's visual quality (e.g., images without artifacts or distortion). The CLIP score also measures text-image alignment, while the Aesthetic Score is a bit different from visual quality. If possible, any human preference metrics on the images' visual quality should be reported.

**Questions:**

- Are the metrics in Figure 3 computed on a test set or a training set?
- The generator is denoted by $g_\theta(x_0|z, c)$. But the authors also write $g_\theta (z|c)$ or $g_\theta (z)$ in some places. They look like a distribution of $z$ instead of $x$. Are they typos or do they mean something else?
- The distance metric $d$ in lines 200-201 is not consistent. It is $d(x, y)$ in some places and $d(x)$ in others.

---

### Meta-Review · Area_Chair_HjXs · 2024-12-19

**Metareview:**

This paper builds on previous efforts to align one-step text-to-image generators with human preferences, using a pre-trained reward model and reference generator. Here the approach adapts RLHF to use a score-based divergence regularization specially formulated in a way that the gradients of the divergence are tractable, which is shown through experiments to improve over previous efforts that relied on the typical KL-divergence formulation. Their model achieves strong human preference scores. Some limitations and future directions are highlighted and discussed.

Strengths:
DI* leads to significant improvements over baselines on human preference scores. The score-based divergence formulation is novel and carefully derived in the paper. Experiments are run at scale with strong opensource models known to the community. The goal is also important for achieving human preferred one-step generation, which is practically relevant. Also the preference alignment does not require new image data for reward tuning. The model defines clear design space choices for reference (diffusion) model and generator architecture.

Weaknesses:
There is a tradeoff between improving human preference alignment and image quality metrics including FID, and reviewers are not convinced that this tradeoff is better handled by DI* over previous methods. Reviewers have some concerns over the complexity of the method and whether it is accomplishing its goals better than the KL-divergence approach of previous works given the above tradeoffs.

Decision reasoning:
While the reviewers and authors engaged in a healthy discussion that significantly improved the paper, reviewers remain borderline on the paper and are concerned that the tradeoff between human preference alignment and image quality scores is problematic and needs to be further examined/addressed via a resubmission. I tend to agree since the main contribution of this paper is an attempted enhancement over DI++, which reviewers remain unconvinced about. I suggest either trying the ‘hacks’ discussed in the paper as ways to ameliorate the tradeoff, or show through further analysis that DI* can achieve a better tradeoff.

**Additional Comments On Reviewer Discussion:**

Reviewers had many concerns that were addressed through thorough experimentation by the authors, who should be commended for this. However, they unanimously remained borderline about the paper contribution.

---

### Decision · Program_Chairs · 2025-01-22

Reject